# SpecBranch: Speculative Decoding via Hybrid Drafting and Rollback-Aware Branch Parallelism

**Yuhao Shen**[1][*]  **Junyi Shen**[2][*]  **Quan Kong**[1]  **Tianyu Liu**[3]  **Yao Lu**[2]  **Cong Wang**[1][†]

[1]Zhejiang University
[2]National University of Singapore
[3]University of Science and Technology of China

```
{riven, quankong, cwang85}@zju.edu.cn
{j1shen, luyao}@comp.nus.edu.sg
tianyu_liu@mail.ustc.edu.cn
```

## Abstract

Speculative decoding (SD) has emerged as a promising technique to accelerate LLM inference by employing a small, efficient draft model to propose draft tokens in advance, and subsequently validating them in parallel with the large target model. However, the existing SD methods still remain fundamentally constrained by their serialized execution, which inevitably causes mutual waiting bubbles between the draft and target models. To address this critical challenge, we draw inspiration from sophisticated branch prediction mechanisms in modern processors and propose a novel framework, **SpecBranch**, to fully unlock branch parallelism in SD. Specifically, we first conduct an in-depth analysis of the potential of branch parallelism in SD, and recognize that the key challenge lies in the intricate trade-offs between parallelization and token rollback. Based on this analysis, we introduce parallel speculative branches to preemptively hedge against likely rejections. Meanwhile, to significantly enhance parallelism, we jointly orchestrate adaptive draft lengths with a hybrid combination of the implicit draft model confidence and explicit reusing of target model features. Extensive experiments conducted across various models and benchmarks show that **SpecBranch** achieves impressive speedups of over $\mathbf{1.8\times \sim 4.5\times}$ against the standard autoregressive decoding and reduces rollback tokens by **50%** for poorly aligned models, while maintaining an identical sampling distribution. Our code is available at https://github.com/Sylvan820/Specbranch.

## 1 Introduction

Recent advances in Large Language Models (LLMs) have revolutionized natural language processing (Achiam et al., 2023; Team et al., 2023; Bai et al., 2023; Guo et al., 2025). However, their real-world deployment faces critical challenges of inference latency due to auto-regressive token-by-token generation, which restricts LLMs to predicting one token at a time, creating a fundamental bottleneck for real-time and large-scale applications.

To address this limitation, Speculative Decoding (SD) has emerged as a promising acceleration paradigm (Leviathan et al., 2023; Chen et al., 2023; Li et al., 2024a). SD uses a small *draft model* to proactively generate candidate tokens, which are then verified in parallel by the large *target model*. By replacing serialized token generation with parallel validation, SD decouples the computational workload from sequence length. However, a critical serial bottleneck still remains. As shown in Fig. 1(a), the draft and target models operate in strict alternation: the draft model idles during target model verification, but the target model cannot process new candidates until the draft model completes

---

[*]Equal contribution.
[†]The Corresponding Authors.

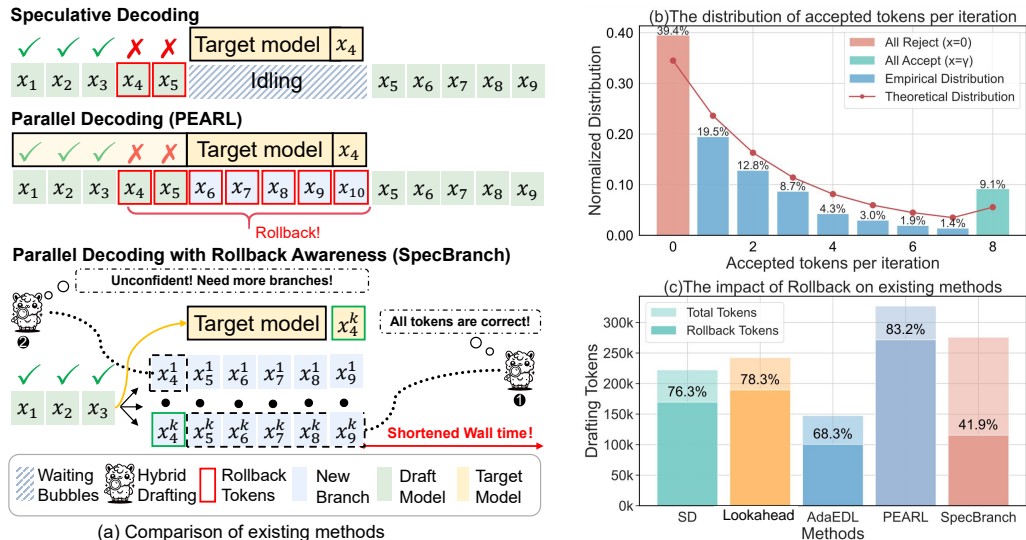

Figure 1: Architectural comparison and empirical analysis of SD frameworks: a) Vanilla SD, Parallel SD (PEARL), and Parallel SD with Rollback Awareness (SpecBranch). When rejection occurs at token $x_4$, PEARL's static pipeline forces verification of those "doomed tokens" $x_5 - x_{10}$; SpecBranch dynamically terminates invalid branches and spawns new branches. b) Distribution of accepted tokens generally follows a truncated geometric distribution with longer token length (Vicuna 68M&13B, $\gamma = 8$); c) Percentage of rollback tokens under different mechanisms.

its proposal. This mutual dependency leads to *pipeline bubbles* (Narayanan et al., 2019) that neither model fully saturates the hardware resources.

Inspired by branch prediction in modern processors (Jiménez & Lin, 2001; Shi et al., 2019), we allow the draft model to *proactively generate speculative branches* concurrently with target model verification. Such a parallel SD paradigm creates a two-stage pipeline in which the draft model token generation overlaps with the target model validation, effectively filling the inherent pipeline bubbles in vanilla SD. Prior works such as PEARL (Liu et al., 2024b) use the target model to verify the first draft token during the drafting phase (pre-verify), and use the draft model to continue generating draft tokens during the verification phase (post-verify). However, unlike lockstep execution from the existing SD that discards tokens only with a local penalty (called "rollback tokens" henceforth), Parallel SD risks **global invalidation** if a token is rejected which causes all subsequent tokens to be rejected, and in turn, stall parallel pipelining. This is exacerbated with longer draft sequence length $\gamma$ since accepted tokens typically follow a truncated geometric distribution as shown in Fig. 1(b). This creates an important trade-off between parallelism and rollback.

Unfortunately, PEARL inadequately addresses these challenges: **1) Pre-verify Rollback.** PEARL overlooks a critical condition for parallel acceleration: the tokens during the verification need to be **All-Accepted**; otherwise, PEARL degenerates to serialized execution and loses its parallel capacity. It verifies only the first token by the target model, while the system remains oblivious to mid-sequence rollback until the parallel verification completes (e.g., when $x_4$ is rejected in Fig. 1(b)). **2) Post-verify Rollback.** The static draft length lacks sufficient awareness of rollback and rejected tokens, which also undermine the benefits of parallelism (shown in Fig. 1(c) with a high percentage of rollback). Consequently, it leads to redundant computation of those "doomed tokens" and makes the target model a bottleneck for processing unnecessary tokens from invalidated branches, despite inevitable rollbacks. This is exacerbated in resource-constrained systems due to misalignment between parameter-imbalanced draft and target models (68M draft&13B target).

Although recent dynamic drafting methods, categorized as implicit (confidence-driven early stopping (Li et al., 2024b; Liu et al., 2024a; Agrawal et al., 2024; Zhang et al., 2023)) or explicit (feature-based sequence modeling (Zhang et al., 2024)) partially mitigate the rollbacks, they face practical challenges from per-task threshold tuning, error compounding and low prediction accuracy. To this end, we propose **SpecBranch** with the following contributions:

| Methods | Parallel Drafting | Model-Training-free | Draft Structure Modeling | Speedup |
|---|:---:|:---:|:---:|:---:|
| Kangaroo (Liu et al., 2024a) | ✗ | ✗ | Implicit (Confidence) | - |
| EAGLE-2 (Li et al., 2024b) | ✗ | ✗ | Implicit (Confidence) | - |
| AdaEAGLE (Zhang et al., 2024) | ✗ | ✗ | Explicit (Feature) | - |
| Lookahead Decoding (Fu et al., 2024) | ✗ | ✓ | None | 1.1×∼1.8× |
| AdaEDL (Agrawal et al., 2024) | ✗ | ✓ | Implicit (Entropy) | 1.4×∼3.0× |
| PEARL (Liu et al., 2024b) | ✓ | ✓ | None (Chunk-level) | 1.6×∼4.2× |
| **SpecBranch (Ours)** | ✓ | ✓ | **Hybrid (Token-level)** | **1.8×∼4.5×** |

Table 1: Comparison of SpecBranch with the existing SD methods. SpecBranch is the first parallel framework with hybrid drafting structures that does not require additional training of draft models.

✧ **Branch-Parallel Architecture**: We first establish theoretical models to quantify ideal parallel speculation and extend it to consider rollback penalties in practice. Guided by these insights, we propose a novel *branch resampling* mechanism that introduces parallel speculative branches to preemptively hedge against likely rejections, while preserving the original sample distribution.

✧ **Hybrid Adaptive Drafting**: Based on extensive empirical analysis of adaptive draft structures, we are the first to unify the implicit (draft model confidence) and explicit (target model feature) methods into a hybrid framework that dynamically optimizes draft lengths. This effectively reduces the percentage of rollback and improves parallel efficiency.

✧ **Extensive Evaluation and Discussion**: We conduct extensive experiments across various models and tasks, demonstrating that SpecBranch consistently achieves a **1.8×** to **4.5×** speedup without draft-model training and reduces rollback tokens by **50%** for poorly aligned draft/target models.

## 2 RELATED WORK

**Speculative Decoding**   While SD has demonstrated significant acceleration and lossless generalization, increasing the acceptance rate of draft tokens by the target model remains a critical challenge. Existing approaches rely on training-based (draft model) and training-free methods to align the draft and target models. For instance, Medusa introduces auxiliary decoding heads Cai et al. (2024), while EAGLE Li et al. (2024a) and Glide Du et al. (2024) reuse target model information to enhance accuracy. SpecInfer uses tree-based attention to verify multiple draft candidates to improve acceptance rates (Chen et al., 2023). On the other hand, training-free methods such as Lookahead decoding adopt a trajectory caching mechanism to store $n$-gram generation histories as draft candidates (Fu et al., 2024). However, all these methods follow a sequential *draft-then-verify* paradigm, which is fundamentally limited by the mutual waiting bottleneck. PEARL (Liu et al., 2024b) introduces a parallel framework that verifies the first draft token while allowing the draft model to simultaneously generate additional tokens during verification. However, it overlooks the impact of rollback when verification fails, which negates the benefit of parallelism if not properly addressed.

**Dynamic Drafting Structures**   Dynamic drafting structure is an effective approach to optimizing SD. It adapts to the draft sequence length or tree configuration (e.g., depth, width, shape) based on contextual speculation. Current methods to model drafting boundaries fall into two categories as illustrated in Table 1: *implicit* and *explicit*. Implicit methods rely on output distribution metrics (e.g., confidence, entropy) to dynamically terminate drafts (Li et al., 2024b; Liu et al., 2024a; Agrawal et al., 2024; Zhang et al., 2023). However, these require manually tuned thresholds and struggle to balance flexibility with computational overhead, particularly in trainable-head variants (Huang et al., 2024; Mamou et al., 2024), which require extra time to predict tokens individually. By eliminating per-token prediction, explicit methods such as AdaEAGLE (Zhang et al., 2024) directly estimate draft lengths based on target model features. Unfortunately, the prediction accuracy declines sharply when the estimation sequence length becomes longer. Essentially, none of these works addresses rollback, a key bottleneck where the draft model wastes tokens and stalls the parallelism. This work proposes a hybrid framework to combine the implicit confidence-based termination with explicit sequence modeling that reduces rollback substantially and improves parallel efficiency.

## 3 PRELIMINARIES

**Notations** We define the *draft model* as $M_q$ and the *target model* as $M_p$. Given a prefix $\mathbf{X}_{1:j} = (x_1, \cdots, x_j)$, $q(\cdot)$ and $p(\cdot)$ denote the probability distributions of the draft and target models, respectively. The speed ratio $c = T_p/T_q$ quantifies the relative latency. Token generation maps $\mathbf{X}_{1:j}$ to embeddings $E_{1:j}$, which transforms to latent features $F_{1:j}$ and generates the next-token distribution $p_{j+1}$ from the final feature vector $f_j$.

**Speculative Decoding** Speculative decoding accelerates autoregressive generation through parallel token verification. The draft model $M_q$ proposes $\gamma$ candidate tokens $\tilde{\mathbf{X}}_{1:\gamma}$ with probabilities $\{q(x_i|\mathbf{X}_{1:i-1})\}_{i=1}^{\gamma}$. The target model $M_p$ computes true probabilities in one forward pass. The acceptance probability for each candidate $x_i$ is $\beta(t_i) = \min\left(1, \frac{p(t_i|\mathbf{X}_{1:i-1})}{q(t_i|\mathbf{X}_{1:i-1})}\right)$. We employ the Match$(p(t_i|\mathbf{X}_{1:i-1}), q(t_i|\mathbf{X}_{1:i-1}))$ function (Zhao et al., 2024) to represent the verification process, which identifies the set of accepted or newly sampled tokens. If $x_i$ is rejected, subsequent candidates $\tilde{\mathbf{X}}_{i+1:\gamma}$ are discarded, and a token is resampled from $\mathrm{norm}(\max(0, p(x_i) - q(x_i)))$; if all $\gamma$ tokens are accepted, an additional token is sampled from $p(t_{\gamma+1})$ (Leviathan et al., 2023).

## 4 ANALYSIS OF PARALLEL DECODING

### 4.1 THEORETICAL SPEEDUP

We first quantify the theoretical speedup of parallel SD under different circumstances. The draft model generates $\gamma$ candidate tokens $\mathbf{X}_{1:\gamma}$, which are verified by the target model. Let $T_q = t$ denote the draft model's per-token generation time, and $T_p = ct$ be the target model's verification time. Under full acceptance of $\gamma$ tokens, the baseline SD achieves $T_{\mathrm{SD}} = \frac{\gamma \cdot T_q + T_p}{\gamma+1} = \frac{\gamma+c}{\gamma+1} \cdot t$.

**Parallel SD (Ideal)** Under the ideal full acceptance condition, the theoretical per-token latency with parallel decoding (shown in Fig. 1(a)) can be derived as:

$$T_{\mathrm{PSD}} = \frac{\max(\gamma t, ct)}{\gamma} = \begin{cases} t, & \gamma \geq c \\ \frac{c}{\gamma}t, & \gamma < c \end{cases} \tag{1}$$

Then the speedup ratio from SD is, $T_{\mathrm{SD}}/T_{\mathrm{PSD}} = \frac{\gamma+c}{\gamma+1}$ or $\frac{\gamma+c}{\gamma+1}\frac{c}{\gamma}$. When $\gamma \approx c$ and $c \gg 1$, PD achieves an optimal $2\times$ speedup against SD. For autoregressive decoding with $\gamma \approx c$, PD represents $c\times$ speedup. Nevertheless, in practice, the actual performance depends on the draft acceptance rate, which has been largely overlooked in the prior work (Liu et al., 2024b).

**Parallel SD (with Rollback)** Recall $\beta$ as the acceptance rate and assume $\beta$s are i.i.d. with $\alpha = \mathbb{E}(\beta)$ as the expected acceptance rate (how well $M_q$ approximates $M_p$). When $k \leq \gamma$, the draft accepted length can be approximated by a truncated geometric distribution (Leviathan et al., 2023) (shown in Fig. 1(b)), detailed in Appendix F.6),

$$P(X = k) = (1-\alpha) \cdot \alpha^k \cdot \mathbb{I}(k < \gamma) + \alpha^\gamma \cdot \mathbb{I}(k = \gamma) \tag{2}$$

where $\alpha^\gamma$ is the probability of *full acceptance* and $1 - \alpha^\gamma$ is the probability of *rollback*. The rollback penalty becomes severe when the draft model has limited capacity, which would cause the subsequent tokens to be discarded and revert parallelism back to serialized execution.

**Theorem 1 (Latency under Rollback).** The per-token latency of parallel SD under rollback is,

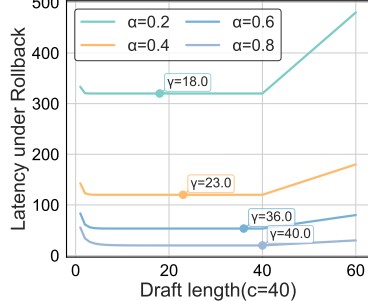

Figure 2: Latency under rollback (Theorem 1). For different $\alpha$, the minimum values are presented (the curves have very mild slopes).

$$T_{\mathrm{PSD}_r} = \frac{2 \cdot \max(\gamma t, ct)}{(1+\alpha^\gamma) \cdot \frac{\alpha(1-\alpha^\gamma)}{1-\alpha}} = \begin{cases} \frac{2ct(1-\alpha)}{\alpha(1+\alpha^\gamma)(1-\alpha^\gamma)}, & \gamma \leq c \\ \frac{2\gamma t(1-\alpha)}{\alpha(1+\alpha^\gamma)(1-\alpha^\gamma)}, & \gamma > c \end{cases} \tag{3}$$

We defer the proofs to Appendix B. As visualized in Fig. 2, the minimum latency occurs at the segment of $\gamma \leq c$, which theoretically validates the trade-off between parallelism and rollback: while

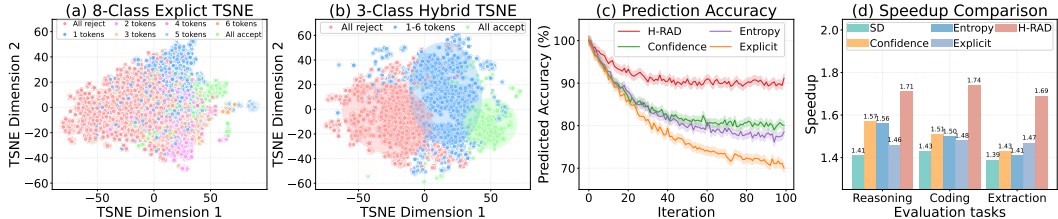

Figure 3: Empirical results of different drafting length estimation strategies: (a,b) comparison of T-SNE visualization for the explicit and the proposed hybrid methods from the MLP activations; (c) both implicit and explicit drafting structures have limitations of low prediction accuracy of the accepted draft length; (d) impact of different drafting schemes on acceleration potentials.

small $\gamma$ underutilizes parallel resources, further increasing $\gamma$ beyond the minimum value leads to diminishing returns due to rollback accumulation. This trade-off is $\alpha$-dependent: for well-aligned models ($\alpha \to 1$), a larger $\gamma$ enjoys parallelism but for misaligned/capacity-constrained draft models $\alpha \leq 0.5$, the penalties from rollback dominate. Though insightful, this theoretical analysis only reflects the statistical properties rather than run-time dynamics. The actual accepted draft length is context-dependent and varies substantially across different iterations (Zhang et al., 2024) (detailed by Fig. 17 in Appendix), which necessitates adaptive control of $\gamma$ rather than static configurations.

## 4.2 ANALYSIS OF ADAPTIVE DRAFT STRUCTURES

Thus, the next question is: "How to optimize draft structures to balance parallelism and rollback?" To answer this, we compare the implicit and explicit methods on LLaMA 68M&7B across the MT-Bench datasets (Zheng et al., 2023) for dialogue tasks. Implicit methods evaluate the confidence $\max_{x_i} q(x_i)$ (Du et al., 2024) or entropy $1 - \sqrt{\lambda H(x_i)}$ (Agrawal et al., 2024) (positively correlated with acceptance rate) against pre-determined thresholds $\epsilon$, but the optimal $\epsilon$ are difficult to find across different tasks, models, temperatures (detailed in Appendix F.8). Additionally, token-level predictions would cause the error-compounding effect with higher instability across different tasks. On the other hand, explicit methods like AdaEAGLE (Zhang et al., 2024) predict accepted length $\gamma$ directly using target model features. Unfortunately, the discriminative power of the explicit methods also declines with the increase of accepted length as validated by the visually inseparable clusters in Fig. 3(a). As a result, the explicit method results in lower prediction accuracy than the implicit method as indicated in Fig. 3(c), despite less overall variance of the ultimate speedup across different tasks (Fig. 3(d)). A partial reason for this difficulty is due to the imbalanced geometric distribution of accepted lengths and the limited contextual information from a single feature layer adopted by (Zhang et al., 2024). Based on these insights, we introduce hybrid rollback-aware branch parallelism.

## 5 SPECBRANCH: HYBRID ROLLBACK-AWARE BRANCH PARALLELISM

To address the intertwined challenges from pipeline bubbles and rollback cost in SD, we present SpecBranch with two novel components: 1) *Hybrid Rollback-Aware Draft Structures (H-RAD)*, which combine the draft model confidence early stopping and target model feature reuse for adaptive draft lengths; 2) *Branch Resampling*, a parallel drafting-verification mechanism that eliminates sequential bottlenecks through context-aware parallelism. We provide a profiling example in Appendix C.

### 5.1 H-RAD: HYBRID ROLLBACK-AWARE DRAFT STRUCTURE

As illustrated by Fig. 4 (Case 1), H-RAD predicts the optimal draft lengths before branch resampling in the draft stage. Unlike PEARL's limited pre-verification of only the first token by target model (Liu et al., 2024b), we leverage the insights from the truncated geometric distribution of accepted tokens in Fig. 1(b) to build a hybrid and lightweight predictor to reduce rollback.

**Hybrid Drafting Length Prediction** Given a prefix, our goal is to accurately predict the draft length $\gamma \leq c$. Since direct regression or multi-class classification of $\gamma$ suffers from low accuracy, the proposed hybrid design reduces the $\gamma$-class classification into a 3 class classification problem. This is inspired by an intriguing bimodal phenomenon that target model features from multiple layers exhibit

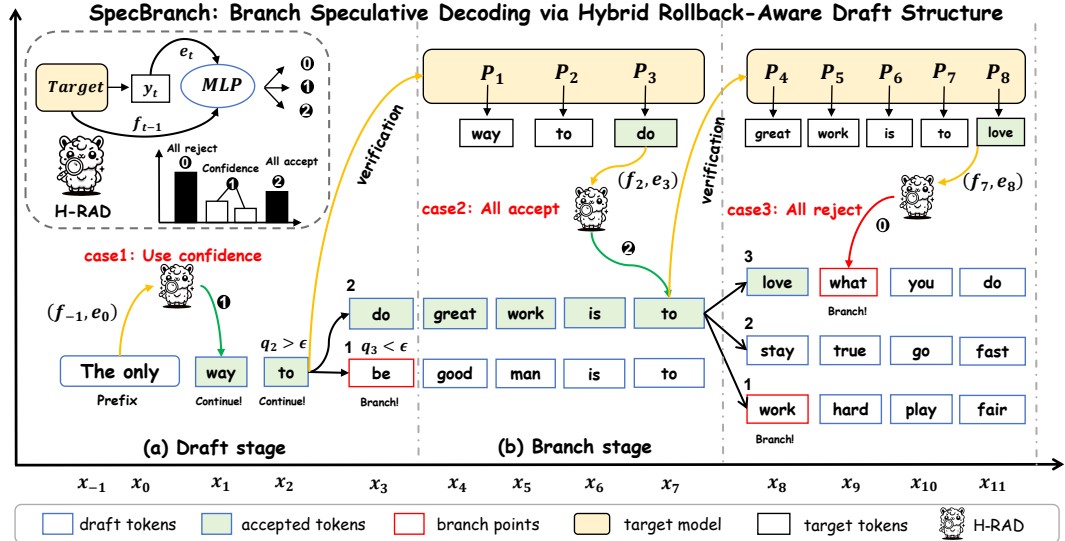

Figure 4: Architecture of SpecBranch. **Case 1 (Use Confidence)**: H-RAD outputs $s_t = 1$, indicating the branch point is determined by the draft model's confidence. **Case 2 (All Accept)**: H-RAD outputs $s_t = 2$, predicting that all tokens should be retained, and the branch point is the next round's first token, *'work'*. **Case 3 (All Reject)**: H-RAD outputs $s_t = 0$, predicting that no tokens should be retained, and the branch point is the first token of this round, *'what'*.

strong separability for the *fully accepted* and *rejected* cases, and the rest, intermediate cases can be resolved by the implicit approach. As illustrated in Fig. 3(b), the hybrid method provides a more separable clustering compared to the overlapping distributions. Thus, we learn a lightweight MLP,

$$\mathbf{z}_t = \text{Concat}(f_{t-1}, \mathbf{e}_t) = \text{Concat}\left(\mathbf{h}_{t-1}^1, \cdots, \mathbf{h}_{t-1}^K, \mathbf{e}_t\right) \in \mathbb{R}^{K \cdot L \cdot D_{\text{layer}} + D_{\text{emb}}}, \quad (4)$$

$$s_t = \arg\max(\text{Softmax}(\text{MLP}(\mathbf{z}_t))) \in \{0, 1, 2\}. \quad (5)$$

where we extract $K$ hidden states $h_{t-1}$ from the target model's last $K$ layers and concatenate them with the new token embedding $e_t$ to $(f_{t-1}, \mathbf{e}_t)$. To capture richer context than single-layer approaches (Li et al., 2024a; Zhang et al., 2024), our method uses multiple layers for better length prediction. The output $s_t$ initiates a new hybrid drafting strategy $\mathcal{H}_t$ with three classes,

$$\mathcal{H}_t = \begin{cases} \emptyset & \text{if } s_t = 0 \quad \text{(Hard signal: All Reject)}, \\ \{x \in \mathbf{X}_{1:\gamma} \mid q(x) > \epsilon\} & \text{if } s_t = 1 \quad \text{(Soft signal: Confidence)}, \\ \mathbf{X}_{1:\gamma} & \text{if } s_t = 2 \quad \text{(Hard signal: All Accept)}. \end{cases} \quad (6)$$

$\mathcal{H}_t$ yields hybrid decisions of $s_0$ and $s_2$ as *hard signals*, and uses draft model confidence $q(x)$ for the intermediate *soft signal* $s_1$. Here, the hard and soft signals refer to direct and pending decisions, respectively. According to the empirical distribution, most tokens are handled by hard signals at once (all accept or reject), while a small fraction is verified by soft signals later. Such a hybrid approach improves the prediction accuracy of the explicit method as well as reduces the compounding errors from the implicit methods. H-RAD predicts the draft length and branch points, providing a dynamic decision for branch resampling, as described in Section 5.2.

## 5.2 BRANCH RESAMPLING: PARALLEL DRAFTING DURING VERIFICATION

**Branch Resampling** H-RAD identifies unconfident tokens as branch points through the hybrid strategy $\mathcal{H}_t$ in three cases as shown in Fig. 4. Then it splits the draft sequence at branch point $x_b$ and sends the prefix $\mathbf{X}_{1:b-1}$ to the target model to verify. Meanwhile, it also spawns $k$ parallel branches via Top-$k$ resampling from the draft model's confidence distribution $q(x_b)$. For example, as illustrated by Fig. 4 case 1, in the sequence *'The only way to be'*, H-RAD outputs $s_t = 1$ to use confidence and the token *'be'* ($q_3 < \epsilon$) triggers a new branch *'do'* with *'way to'* to be verified by the target model,

$$\mathcal{B} = \text{TopK}\left(q(x_b), k\right) = \left\{x_b^1, x_b^2, \ldots, x_b^k\right\}, \quad \text{where } k = \max\left(1, \lfloor k_{\max} \cdot (1 - q(x_b)) \rfloor\right), \quad (7)$$

in which $k$ is adaptively controlled and scales inversely with $x_b$'s confidence $q(x_b)$. Lower confidence in $x_b$ indicates a lower acceptance rate (Du et al., 2024), thus spawning more branches to help SD hedge against likely rejections. Each branch $x_b^i \in \mathcal{B}$ generates subsequent tokens independently using the shared KV-Cache of the prefix $\mathbf{X}_{1:b-1}$ and branch token $x_b^i$ to avoid redundant computation:

$$\mathbf{X}_{\text{branch}}^i = \mathbf{X}_{1:b-1} \oplus x_b^i \oplus M_q \left( \text{KV-Cache}(\mathbf{X}_{1:b-1} \oplus x_b^i) \right). \tag{8}$$

The maximum draft length per branch is constrained by the draft/target model speed ratio $c$, ensuring simultaneous verification and drafting to eliminate pipeline bubbles.

**Branch Verification**    During token drafting, the target model $M_p$ concurrently verifies the last round tokens $\mathbf{X}_{1:b-1}$ using $\text{Match}(p(x_i|\mathbf{X}_{1:i-1}), q(x_i|\mathbf{X}_{1:i-1}))$ (Section 3). If any token $x_i$ is rejected, all subsequent tokens are discarded, and the target model resamples a new token, back to the draft stage. Unlike vanilla SD, if all tokens in $\mathbf{X}_{1:b-1}$ are accepted, we do not need to resample a new token. Instead, we verify the branch point $x_b$ using $p(x_b)$ derived from $x_{b-1}$, apply branch speculative sampling algorithm to sample or adjust the distribution (details in Algorithm 2), consistent with (Li et al., 2024a; Miao et al., 2024), ensuring that the distribution of the output branch point aligns with the target LLM. The branch point verification is formalized as:

$$\mathcal{V} = \text{Match}\big( \underbrace{\big(q(x_b^1) \dots, q(x_b^k)\big)}_{\text{Draft probability}}, \underbrace{\big(p(x_b^1), \dots, p(x_b^k)\big)}_{\text{Target probability}} \big). \tag{9}$$

Here, $\mathcal{V}$ represents the accepted branch point or resampled token after verification. If $\exists\, x_b^i \in \mathcal{V}$, we then remain the corresponding branch, discarding all non-selected branches and their associated KV-Cache. Unlike tree-based methods that spawn branches at every token (expensive KV-Cache growth) (Miao et al., 2024), SpecBranch limits branching to uncertainty points identified by H-RAD and avoids the need for complex tree attention verification. We provide more details in Appendix G.2.

**Posterior Drafting in the Branch Stage**    As shown in Fig. 4 (Case 2/3), the parallel branch stage introduces a potential temporal mismatch between drafting and verification. During the drafting stage, H-RAD predicts the accepted length based on the previous features *before the draft model generates tokens*. However, in the branch stage, the tokens from the previous round have not been verified yet, which means H-RAD cannot immediately access those reliable target model features with sufficient guidance from the history. To address this mismatch, we propose a posterior approach for selecting the retained tokens *after the branch generates tokens*. Once parallel verification is completed, for the remaining $\mathcal{V}$ branch, we use $(f_{t-1}, e_t)$ from the current round as input to H-RAD, and select $\mathcal{H}_t$ after the verification step. This ensures that tokens for the next round are selected based on the most up-to-date context, effectively resolving the mismatch as detailed in Appendix G.2.

## 6    EXPERIMENTS

**Implementation Details**    We evaluate the effectiveness of SpecBranch across diverse LLM configurations, particularly focusing on scenarios where draft models have significantly fewer parameters (68M) and weak alignment with target models (7B-70B), and speedup ratios $c \in [4, 15]$ ($c$ is rounded up to the integer value). This includes LLaMA (Miao et al., 2024) (68M, 7B, $c = 10$), Vicuna (Yang et al., 2024) (68M, 13B, $c = 15$), models with better alignment such as Deepseek-Coder (Guo et al., 2024) (1.3B, 33B, $c = 4$) and LLaMA-3.1 (Grattafiori et al., 2024) (8B, 70B, $c = 5$). We assess SpecBranch across several text generation tasks, including HumanEval (Chen et al., 2021), GSM8K (Cobbe et al., 2021), CNN/DM (Nallapati et al., 2016), and Spec-Bench (widely-adopted six sub-tasks) (Xia et al., 2024b). More details are provided in Appendix E.

**Baseline Methods**    We evaluate against the 4 model training-free methods. **(1) Speculative Decoding (SpS)** (Chen et al., 2023): Standard implementation where a draft model generates $\gamma$ tokens for parallel verification. **(2) AdaEDL** (Agrawal et al., 2024): Early-stopping via entropy-based thresholds to terminate low-probability drafts. **(3) Lookahead Decoding** (Fu et al., 2024): Token-level speculation using cached $n$-gram matches without auxiliary draft models. **(4) PEARL** (Liu et al., 2024b): Pipeline parallelism with pre/post-verification to overlap draft-target model execution.

**Evaluation Metrics**    We report widely used metrics: Mean Accepted Length $M$, Wall-Time Speedup Ratio, and Speed (tokens/sec). In SpecBranch, $M$ represents the continuously accepted length (Liu et al., 2024b). We also introduce a new metric, Rollback Rate (RB), defined as $RB = \frac{\#\text{Rollback tokens}}{\#\text{Total tokens}}$, which quantifies computational waste from invalid drafts, detailed in Appendix E.3.

| Models | Methods | HumanEval | | GSM8K | | CNN/DM | | Speed | Avg. |
|---|---|---|---|---|---|---|---|---|---|
| | | M | Speedup | M | Speedup | M | Speedup | (tokens/s) | Speedup |
| LLaMA 68M&7B | SpS | 2.64 | 1.46× | 3.32 | 1.74× | 2.26 | 1.42× | 63.04 | 1.54× |
| | AdaEDL | 2.49 | 1.54× | 3.25 | 1.89× | 2.19 | 1.46× | 67.21 | 1.63× |
| | Lookahead | 1.48 | 1.31× | 1.96 | 1.71× | 1.45 | 1.25× | 57.63 | 1.42× |
| | PEARL | 2.79 | 1.69× | 3.82 | 1.86× | 2.64 | 1.66× | 71.22 | 1.74× |
| | **SpecBranch** | **3.24** | **2.04×** | **4.46** | **2.12×** | **3.17** | **1.87×** | **82.41** | **2.01×** |
| Vicuna 68M&13B | SpS | 2.87 | 1.79× | 2.54 | 1.56× | 2.07 | 1.45× | 48.79 | 1.60× |
| | AdaEDL | 2.77 | 1.95× | 2.46 | 1.68× | 2.01 | 1.53× | 51.84 | 1.72× |
| | Lookahead | 1.76 | 1.57× | 1.83 | 1.59× | 1.52 | 1.23× | 43.95 | 1.46× |
| | PEARL | 3.11 | 2.02× | 2.83 | 1.61× | 2.89 | 1.68× | 53.31 | 1.77× |
| | **SpecBranch** | **3.69** | **2.47×** | **3.29** | **1.95×** | **3.21** | **1.89×** | **62.57** | **2.10×** |
| Deepseek 1.3B&33B | SpS | 4.45 | 2.16× | 3.85 | 1.86× | 3.91 | 1.96× | 31.38 | 1.99× |
| | AdaEDL | 4.12 | 2.35× | 3.57 | 2.01× | 3.74 | 2.16× | 34.22 | 2.17× |
| | Lookahead | 2.36 | 1.77× | 1.74 | 1.43× | 1.89 | 1.65× | 25.55 | 1.62× |
| | PEARL | 16.97 | 3.39× | 8.28 | 2.78× | 6.45 | 2.63× | 46.17 | 2.93× |
| | **SpecBranch** | **22.52** | **3.71×** | **10.19** | **3.02×** | **7.96** | **2.97×** | **50.94** | **3.23×** |
| LLaMA-3.1 8B&70B | SpS | 5.25 | 2.41× | 5.15 | 2.31× | 5.09 | 2.11× | 16.21 | 2.28× |
| | AdaEDL | 4.96 | 2.55× | 4.97 | 2.37× | 4.85 | 2.19× | 16.91 | 2.37× |
| | Lookahead | - | - | - | - | - | - | - | - |
| | PEARL | 17.28 | 3.75× | 14.33 | 3.35× | 7.51 | 3.04× | 24.03 | 3.38× |
| | **SpecBranch** | **21.74** | **4.02×** | **18.08** | **3.67×** | **9.41** | **3.37×** | **26.27** | **3.69×** |

Table 2: Comparison with existing baselines on HumanEval (Chen et al., 2021), GSM8K (Cobbe et al., 2021) and CNN/DM (Nallapati et al., 2016). "–" indicate incompatibility: baseline implementations (transformers = 4.36.2) conflict with LLaMA 3.1's dependency on ≥ 4.43.0.

| Models | Methods | MT Bench | | QA | | Sum | | Math | | RAG | | Trans | | Avg. |
|---|---|---|---|---|---|---|---|---|---|---|---|---|---|---|
| | | M | Speedup | M | Speedup | M | Speedup | M | Speedup | M | Speedup | M | Speedup | |
| Vicuna | SpS | 2.63 | 1.74× | 2.47 | 1.64× | 2.58 | 1.70× | 2.37 | 1.55× | 2.47 | 1.56× | 2.57 | 1.65× | 1.64× |
| | AdaEDL | 2.50 | 1.80× | 2.43 | 1.67× | 2.64 | 1.72× | 2.31 | 1.62× | 2.21 | 1.57× | 2.45 | 1.75× | 1.69× |
| | Lookahead | 1.56 | 1.31× | 1.41 | 1.23× | 1.49 | 1.25× | 1.71 | 1.46× | 1.38 | 1.15× | 1.32 | 1.10× | 1.25× |
| | PEARL | 2.62 | 1.78× | 2.45 | 1.64× | **2.78** | **1.83×** | 2.63 | 1.67× | 2.61 | 1.66× | 2.89 | 2.05× | 1.77× |
| | **SpecBranch** | **3.11** | **2.09×** | **2.67** | **1.83×** | 2.72 | 1.78× | **2.86** | **1.89×** | **2.83** | **1.86×** | **3.32** | **2.30×** | **1.96×** |
| LLaMA-3.1 | SpS | 4.67 | 2.31× | 4.57 | 2.27× | 5.09 | 1.98× | 5.01 | 2.44× | 5.08 | 2.02× | 5.52 | 2.57× | 2.27× |
| | AdaEDL | 4.31 | 2.43× | 4.23 | 2.30× | 4.83 | 2.05× | 4.94 | 2.46× | 4.86 | 2.13× | 5.24 | 2.65× | 2.34× |
| | Lookahead | - | - | - | - | - | - | - | - | - | - | - | - | - |
| | PEARL | 8.46 | 2.96× | 8.37 | 3.27× | 9.10 | 3.32× | 12.53 | 3.39× | 8.35 | **3.41×** | 12.59 | 4.22× | 3.43× |
| | **SpecBranch** | **10.85** | **3.24×** | **10.59** | **3.45×** | **11.40** | **3.63×** | **15.76** | **3.78×** | **9.16** | 3.40× | **16.64** | **4.51×** | **3.67×** |

Table 3: Comparison with the existing baselines on Spec-Bench (Xia et al., 2024b).

**H-RAD Training and Generalization** The training data for H-RAD pairs the feature vector $z_t$ from Eq. (4) with the corresponding three-class labels $s_t$. We implement a lightweight three-layer MLP with ReLU activation. Training is performed *offline* for 20 epochs and 32 batch size. The training converges within 5 minutes on a single NVIDIA A100 GPU and eliminates the need for costly online fine-tuning. We further conduct experiments demonstrating the cross-task generalization of H-RAD, where its performance only degrades by 5% (Table 8). We provide details in Appendix E.4.

## 6.1 MAIN RESULTS

The main results from Tables 2, 3 and Fig. 5 are explained below: **(I)** On HumanEval, GSM8K, and CNN/DM, SpecBranch shows superior efficiency over prior methods, achieving consistent speedups of $1.9\times$ to $4.0\times$ over vanilla autoregressive decoding. On Spec-Bench, our method also achieves significant acceleration from $1.8\times$ to $4.5\times$ on six diverse sub-tasks, indicating its robustness and versatility. **(II)** For poorly aligned models (LLaMA, Vicuna), rollback dominates parallel acceleration. SpecBranch improves over PEARL by 15%. As shown in Fig. 5, SpecBranch reduces rollback from 66-90% to under 40%, yielding longer generated lengths $M$ compared to the baseline (Tables 2, 3). This validates the capability of H-RAD to terminate those draft paths with ultimate failure. **(III)** For

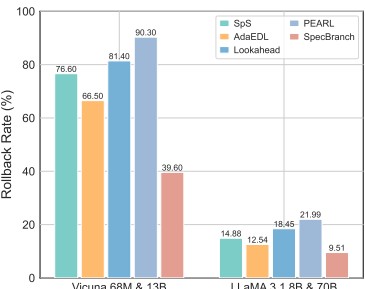

Figure 5: Comparison of Rollback Rates on HumanEval.

well-aligned models (Deepseek, LLaMA-3.1), SpecBranch improves parallelism and resource uti-

lization through branch resampling. E.g., it achieves $3.23\times$ speedup vs. PEARL's $2.93\times$ $(10.2\%)$, with a $4.14\times$ improvement of the average accepted length $M$ against SpS on code generation tasks. It also reduces the rollback rate by $10\%$ and improves speedup by $8\%$ compared to PEARL. These results empirically validate the trade-off in Theorem 1 and more details are provided in Appendix F.

## 6.2 ABLATION STUDIES

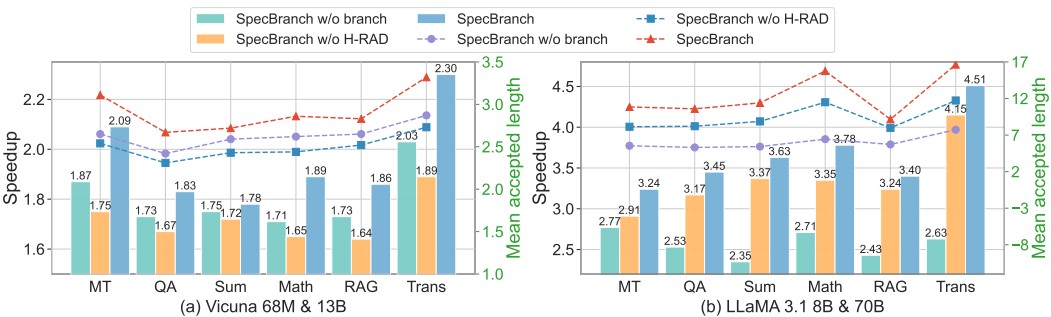

Figure 6: Component Ablation on the Spec-Bench benchmark: (a) for poorly aligned model pairs like Vicuna 68M-13B, H-RAD provides higher contributions; (b) for better-aligned models such as LLaMA-3.1 8B-70B, branch resampling plays a more important role in acceleration.

**Component Analysis** Our ablation studies first isolate the two core components by removing: (1) branch resampling (SpecBranch *w/o branch*), or (2) H-RAD (SpecBranch *w/o H-RAD*). Fig. 6 reveals that the absence of either component degrades performance. H-RAD dominates efficiency gains for misaligned model pairs like Vicuna 68M-13B, which improves the speedup from $1.72\times$ to $1.95\times$. On the other hand, branch resampling contributes more with better-aligned pairs like LLaMA-3.1 8B-70B. Both components offer complementary advantages to reduce rollback and improve parallelism by adapting to different model capacity. Meanwhile, branch parallelism and H-RAD are both modular designs, which are orthogonal to methods like EAGLE (Li et al., 2024a;b) and can be quickly adapted, which we leave for future exploration as detailed in Appendix G.2.

**Hyperparameter Sensitivity** Since H-RAD still entails the confidence threshold from the implicit methods, we evaluate its sensitivity to such threshold and feature layer hyperparameters in Tables 4 and 5 by integrating H-RAD with vanilla SD. Table 4 shows that while the speed for implicit methods drops from 64 to 49 tokens/s as $\epsilon$ increases, H-RAD only decreases from 72 to 67 tokens/s, with less dependence on the threshold. The explicit variant in Table 5 reveals diminishing returns with more feature layers ($K$), that increasing context layers $K$ from 4 to 32 has marginal gains of $1-2$ tokens/s but $8\times$ more memory overhead. Thus, we choose $K = 4$ to balance the speed and memory.

| $\epsilon$ | Implicit(Confidence) | Implicit(Entropy) | Hybrid(H-RAD) |
|---|---|---|---|
| 0.1 | 61.05 | 60.28 | 70.32 |
| **0.2** | **64.26** | **63.03** | **72.15** |
| 0.4 | 61.12 | 59.21 | 71.02 |
| 0.6 | 56.43 | 54.73 | 69.91 |
| 0.8 | 53.21 | 52.29 | 68.62 |
| 0.9 | 49.46 | 48.18 | 67.31 |

| $K$ | HumanEval(coding) | GSM8K(reason) | CNN/DM(sum) |
|---|---|---|---|
| 1 | 62.35 | 63.28 | 52.82 |
| 2 | 64.06 | 76.14 | 57.46 |
| **4** | **72.15** | **79.24** | **63.46** |
| 8 | 73.28 | 80.22 | 63.86 |
| 16 | 73.83 | 81.27 | 64.35 |
| 32 | 74.18 | 81.33 | 64.41 |

Table 4: Results of Stop thresholds $\epsilon$ of LLaMA 68M&7B on HumanEval. (tokens/sec)

Table 5: Results of feature layers $K$ of LLaMA 68M&7B on H-RAD+SD. (tokens/sec)

**Resource Consumption** We further validate SpecBranch's effectiveness across three dimensions of resource consumption. For memory consumption, we test the LLaMA-3.1 models on HumanEval; for energy/time cost, we use poorly-aligned Vicuna on HumanEval. Fig. 7(a) shows that, due to the shared prefix KV-Cache, memory consumption for parallel branches with varying $k$ (from 1 to 16) only has a slight increment to $28\%$ of the baseline model parameters. In SpecBranch, $k$ is dynamically adjusted based on confidence at the branch point with $k_{max}$ typically capped by 6. SpecBranch strategically spawns sparse branch points at high-impact tokens, which improves computational and memory efficiency by avoiding unnecessary branching. By the rollback-aware designs, Fig. 7(b) demonstrates that SpecBranch reduces energy consumption by $43\%$ against PEARL. Fig. 7(c) shows the time cost

for each SpecBranch module, where the H-RAD prediction cost is negligible (0.38% of total latency). SpecBranch eliminates the mutual waiting bubbles between draft and target models with almost identical execution time (30.9 vs 31.4 ms per step). We provide more details in Appendix F.4, G.1.

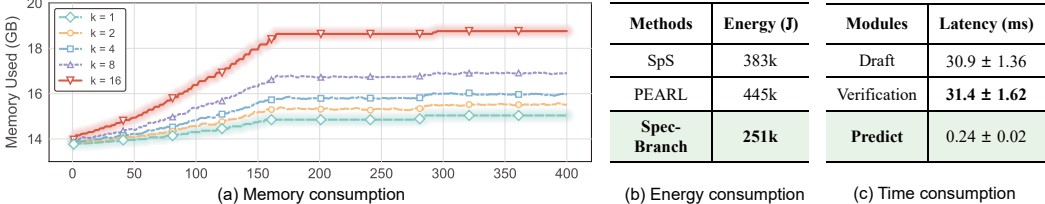

| Methods | Energy (J) | Modules | Latency (ms) |
|---|---|---|---|
| SpS | 383k | Draft | 30.9 ± 1.36 |
| PEARL | 445k | Verification | **31.4 ± 1.62** |
| **Spec-Branch** | **251k** | **Predict** | 0.24 ± 0.02 |

(a) Memory consumption  (b) Energy consumption  (c) Time consumption

Figure 7: Resource consumption of SpecBranch through NVIDIA DCGM. (a) Trace of memory footprint of SpecBranch inference with different number of branches $k$ on HumanEval using LLaMA-3.1 8B-70B; (b)(c) Energy and time cost on HumanEval using Vicuna 68M-13B.

**Branch Length**    We define the max branch length as the default draft length $\gamma$ in each decoding round. Unlike PEARL, which statically fixes $\gamma = c$ (the theoretical speedup ratio) for all model pairs, SpecBranch introduces a rollback-aware parallel SD architecture, optimizing $\gamma$ based on the parallelism–rollback trade-off. As shown in Fig. 8, we evaluate two representative cases on HumanEval: LLaMA 68M/7B (low alignment, $c = 10$) and LLaMA-3.1 8B/70B (high alignment, $c = 5$). For the low-alignment pair, the optimal $\gamma = 8$ falls strictly below $c$, validating the trade-off analysis in Fig. 2. In this regime, the latency penalty from rollbacks dominates parallelism gains, which is a critical nuance PEARL overlooks. Conversely, for high-alignment pairs, the optimal $\gamma$ approaches $c$ as the benefits of parallelism prevail over rollback costs.

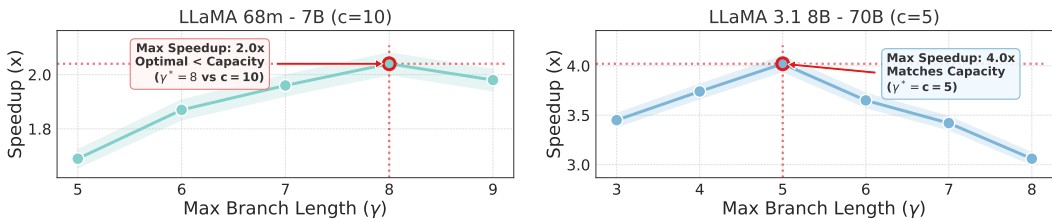

Figure 8: Impact of max branch length $\gamma$ on inference speedup. Unlike PEARL's fixed setting ($\gamma = c$), the optimal $\gamma$ varies by model alignment: (a) for the poorly aligned LLaMA pair ($c = 10$), optimal $\gamma = 8$ falls below $c$ due to dominant rollback costs; (b) for the well-aligned LLaMA-3.1 pair ($c = 5$), optimal $\gamma$ approaches $c$ as parallelism benefits prevail.

**More Discussion**    Due to space, we include more results and discussions in the Appendix, including more comparisons with the tree methods (EAGLE) ( F.1, G.10), high-batch performance ( G.9), $k_{max}$ analysis ( F.2), memory-constrained application ( G.1), tree structure and temporal mismatch ( G.2) and more ablation results ( F.8, F.10).

## 7    CONCLUSION

**Future Work**    Current parallel speculative decoding relies on off-the-shelf draft models, where semantic misalignment often limits verification efficiency. To address this bottleneck, we plan to explore on-policy online distillation (Lu & Lab, 2025) to dynamically adapt draft policies for superior alignment with the target model. Furthermore, we aim to synergize SpecBranch's parallel H-RAD mechanism with advanced feature-level speculative methods (e.g., EAGLE) and deploy these integrated engines into high-performance serving frameworks like vLLM and SGLang, thereby satisfying the rigorous demands of industrial environments.

We propose SpecBranch, a rollback-aware parallel SD framework. By enabling concurrent branch drafting and verification, SpecBranch addresses the bottleneck of serialized execution via two key innovations of branch resampling and hybrid drafting structures. Experiments show that SpecBranch achieves $1.8 \sim 4.5\times$ speedup and reduces computational waste up to 50% for poorly aligned models, while theoretically guaranteeing the preservation of the generated text's distribution.

## ACKNOWLEDGEMENTS

This work is supported by Zhejiang Provincial National Science Foundation of China under Grant No. LZ25F020007 and National Science Foundation of China under grants 62576310, 62394341.

## ETHICS STATEMENT

The datasets used in our experiments are publicly released and labeled through interaction with humans in English. In this process, user privacy is protected, and no personal information is contained in the dataset. The scientific artifacts that we used are available for research with permissive licenses. The use of these artifacts in this paper is consistent with their intended purpose.

## REPRODUCIBILITY STATEMENT

We discuss the experimental settings in Section 6 and Appendix E, including implementation details such as models, datasets, inference setup, and evaluation metrics.

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

# A  PROCEDURES OF SPECBRANCH

---

**Algorithm 1:** Algorithm of SpecBranch

---

**Input:** Prefix $I$, draft model $M_q$, target model $M_p$, max length $L$, target model feature $f$, target model embedding $e$, predicted output $s_t$, confidence threshold $\epsilon$, num branches $B$, max gamma $\gamma_m$, execution mode $E$.

**Output:** Generated sequence $\mathbf{x}$.

$\mathbf{x} \leftarrow I, \gamma \leftarrow \gamma_m, E \leftarrow \texttt{DRAFTING}$

**while** $|\mathbf{x}| < L$ **do**

    **if** $E = DRAFTING$ **then**

        *// Drafting phase: only drafting candidate tokens*

        $\gamma = \texttt{Predictor}(f_{t-1}, e_t, \epsilon, \gamma_m)$

        **for** $i \leftarrow 1$ **to** $\gamma$ **do**

            $q_i \leftarrow M_q\big(\mathbf{x} + [x_1, \ldots, x_{i-1}]\big)$

            $x_i \sim q_i$

        **end**

        $E \leftarrow \texttt{VERIFICATION}$

    *// Verification phase: evaluate branches*

    **for** $b \in \{1, \ldots, B\}$ **do**

        **for** $i \leftarrow \gamma + 1$ **to** $2\gamma$ **do**

            $q_{(b,i)} \leftarrow \texttt{Mask}(M_q\big(\mathbf{x} + [x_{(b,1)}, \ldots, x_{(b,i-1)}]\big))$

            $x_{(b,i)} \sim q_{(b,i)}$

            **if** $q_{(b,i)}[x_{(b,i)}] < \epsilon$ **then**

                **keep the invalid position**

        **end**

    **end**

    $(p_1, \ldots, p_\gamma) \leftarrow \big(M_p(\mathbf{x} + [x_1]), \ldots, M_p(\mathbf{x} + [x_1, \ldots, x_\gamma])\big)$

    Retrieve $(q_1, \ldots, q_\gamma)$ from cache

    **for** $i \leftarrow 1$ **to** $\gamma$ **do**

        $r_i \sim U(0,1)$

    **end**

    $n \leftarrow \min\big(\{\, i - 1 \mid r_i > \frac{p_i[x_i]}{q_i[x_i]} \,\} \cup \{\gamma\}\big)$

    **if** $n = \gamma$ **then**

        *// All drafted tokens accepted*

        **if** $\exists b : \ r_b \leq \frac{p[x_{n+1}]}{q_{(b,n+1)}[x_{n+1}]}$ **then**

            $\mathbf{x} \leftarrow \mathbf{x} + \big[x_{(b^*,n+1)}, \ldots, x_{(b^*,\gamma)}\big]$

            $\gamma \leftarrow \texttt{Predictor}(f_t, e_{t+1}, \epsilon, \gamma_m)$

            $E \leftarrow \texttt{VERIFICATION}$

        **else**

            *// Reject next token — fallback to target*

            $y \sim \mathcal{N}\big(\max(0, p_{n+1} - q_{(b,n+1)})\big)$

            $\mathbf{x} \leftarrow \mathbf{x} + [y]$

            $E \leftarrow \texttt{DRAFTING}$

        **end**

    **else**

        *// Rejection occurred at position $n$*

        $y \sim \mathcal{N}\big(\max(0, p_{n+1} - q_{n+1})\big)$

        $\mathbf{x} \leftarrow \mathbf{x} + [x_1, \ldots, x_n, y]$

        $E \leftarrow \texttt{DRAFTING}$

    **end**

**end**

**return** $\mathbf{x}$

---

# B  PROOF OF THEOREM 1

We re-state Theorem 1 (Section 4.1) here for convenience with the basic assumption from (Leviathan et al., 2023) that the token acceptance is i.i.d. with $\mathbb{P}(\text{accept}) = \alpha$. Recall that the generation time $T_q = t$, the verification time $T_p = ct$, and any rollback would trigger a full retry of $\gamma$ tokens. To prove Theorem 1, we present Lemma 1 on the expected token draft length first.

**Lemma 1 (Expected Draft Accepted length)** *For truncated geometric distribution,*
$X \sim TruncGeo(\alpha, \gamma)$,

$$E[X] = \frac{\alpha(1 - \alpha^\gamma)}{1 - \alpha} \tag{10}$$

*Proof.*

$$E[X] = \sum_{k=0}^{\gamma} k \cdot \mathbb{P}(X = k) = \sum_{k=0}^{\gamma-1} k \cdot (1 - \alpha)\alpha^k + \gamma \cdot \alpha^\gamma$$

Let $S = \sum_{k=0}^{\gamma-1} \alpha^k = \frac{1-\alpha^\gamma}{1-\alpha}$. Take differentiation regarding $\alpha$, we have,

$$\frac{dS}{d\alpha} = \sum_{k=0}^{\gamma-1} k\alpha^{k-1} = \frac{1 - \gamma\alpha^{\gamma-1} + (\gamma - 1)\alpha^\gamma}{(1 - \alpha)^2}$$

$$E[X] = (1 - \alpha)\alpha \cdot \frac{dS}{d\alpha} + \gamma \cdot \alpha^\gamma = \frac{\alpha(1 - \alpha^\gamma)}{1 - \alpha}$$

**Theorem 1 (Latency under Rollback)** *The latency of parallel SD under rollback is,*

$$T_{BD_r} = \frac{2 \cdot \max(\gamma t, ct)}{(1 + \alpha^\gamma) \cdot \frac{\alpha(1-\alpha^\gamma)}{1-\alpha}} = \begin{cases} \dfrac{2ct(1 - \alpha)}{\alpha(1 + \alpha^\gamma)(1 - \alpha^\gamma)}, & \gamma \leq c \\ \dfrac{2\gamma t(1 - \alpha)}{\alpha(1 + \alpha^\gamma)(1 - \alpha^\gamma)}, & \gamma > c \end{cases}$$

*Proof.* Define the acceptance vector $\omega = (\omega_1, \ldots, \omega_\gamma) \in \{0, 1\}^\gamma$, where $\omega_i = 1$ if and only if token $i$ is accepted. The accepted token count is,

$$X = \sum_{i=1}^{\gamma} \omega_i, \quad \mathbb{P}(\omega_i = 1) = \alpha \text{ (i.i.d.)}.$$

To compute the total number of tokens with retry, define two rounds of: 1) $\gamma$ tokens (accepted if $\omega = \mathbf{1}$); 2) Retry if Round 1 fails, which yields $\mathbb{E}[X]$ tokens. Thus, the total expectation is

$$E_{\text{total}} = \alpha^\gamma(\gamma + E[X]) + (1 - \alpha^\gamma)\frac{(E[X] - \gamma\alpha^\gamma)}{1 - \alpha^\gamma} = (1 + \alpha^\gamma) \cdot E[X] \tag{1}$$

This implies that Parallel SD (with Rollback) achieves an acceleration factor of $(1 + \alpha^\gamma)\times$ compared to the vanilla SD (with Rollback). As $\alpha$ approaches 1, the acceleration ratio reaches $2\times$, matching the acceleration of the Ideal Parallel SD in Eq. (1). Thus, the total time for the two rounds (parallel generation/verification):

$$T_{\text{total}} = 2 \cdot \max(\gamma t, ct) \tag{2}$$

$$T_{\text{PSD}_r} = \frac{T_{\text{total}}}{E_{\text{total}}} = \frac{2 \cdot \max(\gamma t, ct)}{(1 + \alpha^\gamma) \cdot \frac{\alpha(1-\alpha^\gamma)}{1-\alpha}} \tag{3}$$

For different cases of length $\gamma$, we have,

Case 1: $\gamma \leq c$ (Verification time $ct > \gamma t$):

$$T_{\text{PSD}_r} = \frac{2ct}{(1 + \alpha^\gamma) \cdot \frac{\alpha(1-\alpha^\gamma)}{1-\alpha}} = \frac{2ct(1 - \alpha)}{\alpha(1 + \alpha^\gamma)(1 - \alpha^\gamma)}.$$

Case 2: $\gamma > c$ (Generation time $\gamma t > ct$):

$$T_{\text{PSD}_r} = \frac{2\gamma t}{(1 + \alpha^\gamma) \cdot \frac{\alpha(1-\alpha^\gamma)}{1-\alpha}} = \frac{2\gamma t(1 - \alpha)}{\alpha(1 + \alpha^\gamma)(1 - \alpha^\gamma)}.$$

(Simplified using $\gamma t = ct \cdot \frac{\gamma}{c} > ct$). This completes the proof of Theorem 1.

## C  PROFILING EXAMPLE

We have illustrated the main process of SpecBranch in Fig. 4. Here, we provide a more detailed step-by-step profiling example of SpecBranch with an input prefix 'the only' in Fig. 9.

| Step | Prefix | Input Signals | Draft model Output | Target model Output | Judgement Tips |
|---|---|---|---|---|---|
| 1 | The only | draft stage (predict) 

 st =1 
 q(be) < eps | way to **be** | \No | In the draft stage, H-RAD outputs st = 1 and uses confidence to decide the branch point before generating tokens. If q(be) < eps, the branch is in 'be'. |
| 2 | The only | branch stage (posterior) 

 st=2 
 all accept | (1) **be** good man is to 
 (2) do great work is to | way to do | In the branch stage, 'be' and 'do' continue as branches. After verification, 'do' is kept, and H-RAD outputs st = 2, retaining all tokens for the next verification. |
| 3 | The only way to do | branch stage (posterior) 

 st=0 
 all reject | (1) **work** hard play fair 
 (2) stay true go fast 
 (3) love **what** you do | great work is to love | In the branch stage, 'work' starts branchs, 'stay' 'love'. After verification, 'love' is retained, and H-RAD outputs st = 0, discarding all tokens while branching to 'what'. |
| 4 | The only way to do great work is to love | branch stage (posterior) 

 st=1 
 q(devote) < eps | (1) **what** you do throughout the 
 (2) the efforts you **devote** during | the | In the branch stage, 'what' branches to 'the'. After verification, 'the' remains, and H-RAD outputs st = 1, using confidence and q(devote) < eps to branch into 'devote'. |
| 5 | The only way to do great work is to love the | branch stage (posterior) 

 st=2 
 all accept | (1) **devote** during the whole research 
 (2) put into the whole ~~project~~ | the efforts you put | In the branch stage, 'devote' branches to 'put'. After verification, 'put' is kept, and H-RAD outputs st = 2, retaining all tokens for the next verification. |
| 6 | The only way to do great work is to love the efforts you put | branch stage (posterior) 

 **Rollback!** | (1) ~~but not the fame~~ 
 (2) ~~and the insight you~~ | into the whole **research** | In the branch stage, 'but' branches to 'and'. After verification, **project is rejected**, and a rollback occurs to 'whole' concatenated with 'research'. |
| 7 | The only way to do great work is to love the efforts you put into the whole research | draft stage (predict) 

 st=2 
 all accept | but not the successes | \No | Once a token is rejected by the target model (either at the branch point or as a normal token), the process returns to the draft stage and repeats. |

Figure 9: A step-by-step profiling example of SpecBranch with an input prefix 'the only'.

The cyan text represents the stage and the red text indicates branch points or rejected tokens. Underlined text signifies the tokens retained after each round of posterior drafting in the branch stage, which are sent to the target model for verification. The dashed lines represent the tokens discarded during rollback. We give some explanations about the whole process step by step.

1) At step 1, we extract features from the prefix 'The only' and input it to H-RAD (assuming the preceding context is already established), which outputs the generation strategy $s_t = 1$. It means that we use the implicit confidence $q(x)$ to determine when to begin branching. Then, 'The only' is input to the draft model, and the target model does not operate during this stage. The draft model generates 'way to' until $q(\text{be}) < \epsilon$, at which point we branch at 'be' and send 'way to' to the target model for parallel verification, transitioning from the draft stage to the branch stage.

2) At step 2, 'be' and 'do' are generated in parallel as two separate branches, while 'way to' is verified by the target model. Once both drafting and verification are completed, the target model outputs 'way to do,' indicating that the second branch 'do' is selected and the 'be' branch is discarded. This allows SpecBranch to successfully avoid the rollback of 'be' through H-RAD prediction and branch resampling. Then, H-RAD reuses the feature output $s_t = 2$, indicating that all tokens have a high acceptance probability, and 'great work is to' is sent for verification in the next round.

3) At step 3, since $s_t = 2$, 'word' is the first token of this round and is considered as an unconfident token, prompting the generation of new branches 'stay' and 'love'. After the target model verifies and accepts the sequence 'great work is to,' it also outputs 'love' to match the branch. By generating a new branch, the rejection of 'work' is avoided. Then, H-RAD reuses the feature output $s_t = 0$, indicating that all tokens have a low acceptance probability, and only 'love' is sent to the target model to match the branch point 'what'.

4) At step 4, 'what' branches to 'the' and generates tokens in parallel. Then, the target model outputs 'the', meaning 'what' is rejected and the 'the' branch is retained, thus avoiding the rollback of 'what'. H-RAD and branch resampling combine to improve token rollback handling, while PEARL suffers from token rollback. Then, H-RAD outputs $s_t = 1$, meaning we need to use confidence to determine the branch point 'devote', and send 'the efforts you' to the target model for the next round of verification.

5) At step 5, similarly, SpecBranch combats the unconfident token 'devote' through implicit confidence early stopping and branch resampling, improving parallel efficiency. Once 'the efforts you put' is accepted, H-RAD outputs $s_t = 2$ to retain all tokens generated by the 'put' branch.

6) At step 6, we provide a rollback case. In the new round, the first token 'but' branches to 'and', drafting tokens. However, the target model rejects the previous round's token 'project' and resamples a new token 'research'. This means that all the tokens generated in parallel need to be discarded, and we return to 'whole' and concatenate with 'research'. Meanwhile, if none of the branch points match the output of the target model, this is also treated as a rollback.

7) At step 7, once a token is rejected by the target model (either at the branch point or as a normal token), the process returns to the draft stage and repeats the above process of draft branch in parallel. For those "doomed tokens" that cannot be avoided, they have to be rolled back to the draft stage. This completes the feedback loop.

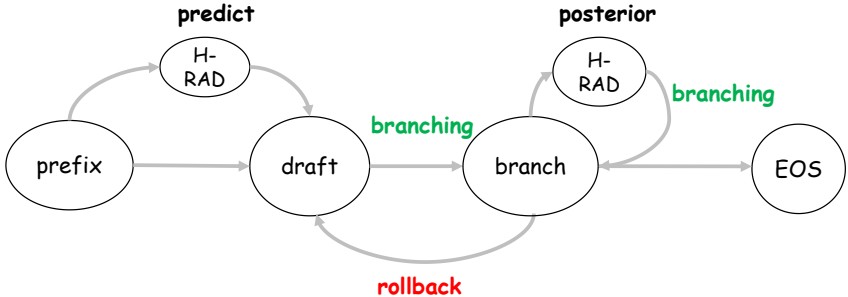

Figure 10: Illustration of the SpecBranch state transitions.

Meanwhile, to better illustrate the SpecBranch workflow, we present a simplified stage-transition logical loop in Fig. 10. Given a prefix, if target-model features are available, H-RAD reuses these features to predict the draft length during the draft stage. The transition from the draft stage to the branch stage occurs when a branch point is selected, which can be one of three scenarios: (1) $s_t = 0$ (all-reject), indicating that the branch point is the first token of this round; (2) $s_t = 1$, where confidence is used to decide the branch point; and (3) $s_t = 2$ (all-accept), indicating that the branch

point is the first token of the next round. Each scenario triggers a distinct branching to ensure correct overlap between drafting and verification.

Similarly, in the branch stage, H-RAD uses a posterior approach to select the retained tokens, leading to three branching scenarios for parallel execution. Once a token is rejected by the target model (either at the branch point or as a regular token), the branch stage reverts to the draft stage and repeats the draft-branch process in parallel, thus maintaining the logical loop.

## D  BRANCH SPECULATIVE SAMPLING

As discussed in Section 5.2, we verify the branch point $x_b$ using $p(x_b)$ derived from $x_{b-1}$, and apply branch speculative sampling algorithms to sample or adjust the distribution of candidate branches. The branch point verification is formalized as:

$$\mathcal{V} = \text{Match}\Big(\underbrace{\big(q(x_b^1)\dots,q(x_b^k)\big)}_{\text{Draft probability}}, \underbrace{\big(p(x_b^1),\dots,p(x_b^k)\big)}_{\text{Target probability}}\Big)\Big\}. \tag{11}$$

Single-round speculative sampling relies on a chain-structured draft, whereas SpecBranch adopts a branch structure. Inspired by (Li et al., 2024a; Miao et al., 2024), SpecBranch introduces Branch Speculative Sampling. Specifically, given the target model's distribution $p(x_b)$ for the branch point, we perform speculative sampling token-by-token on the top-$k$ branch point candidates. For each candidate $x_b^i$, we evaluate it against the criterion $r_b < \frac{p(x_b^i)}{q(x_b^i)}$. Once any branch point token $x_b^i$ is verified and accepted, the corresponding branch is selected and retained. In non-acceptance scenarios, branch speculative sampling recursively invokes single-round speculative sampling, instead of retaining the original naive sampling. The pseudocode corresponding to Branch Speculative Sampling is detailed in Algorithm 2.

---

**Algorithm 2:** Branch Speculative Sampling

---

**Input:** Branch points target model distribution $p(x_b)$, top-$k$ branch points $x_b^i$ and distributions $q(x_b^i)$ for each $i$ from 1 to $k$, where $x_b^i$ is sampled from $q(x_b^i)$,
**Output:** a sample $x_b \sim p(x_b)$;
$i \leftarrow 1$
**for** $i \leq k$ **do**
    $r_b \leftarrow U(0,1)$
    **if** $r_b < p(x_b^i)/q(x_b^i)$ **then**
        **Return** $x_b^i$
    **end if**
    $p(x_b) \leftarrow \texttt{norm}(\max(0, p(x_b) - q(x_b^i)))$
    $i \leftarrow i + 1$
**end for**
Sample a new token $x_b \sim p(x_b)$
**Return** $x_b$

---

Unlike tree-based methods that spawn branches at every token (expensive KV-Cache growth) (Miao et al., 2024), SpecBranch limits branching to uncertainty points identified by H-RAD and avoids the need for complex tree attention mask verification. This means SpecBranch only applies branch speculative sampling at branch points, while employing the typical speculative sampling for subsequent draft tokens. This ensures an **identical sampling distribution of the target model**, enabling lossless acceleration while also significantly reducing deployment complexity and computational overhead.

Our main experiments are conducted **under greedy decoding conditions** that are consistent across all baselines i.e., target model temperature $= 0$. To further verify the lossless nature of our method, we perform additional experiments on GSM8K across different temperatures to evaluate accuracy. As shown in the Table 6, despite a slight variation for Vicuna, SpecBranch achieves identical accuracy with the vanilla autoregressive decoding across various model pairs and temperatures. Branch Speculative Sampling theoretically guarantees the original output distribution of the target LLM.

| Models | Methods | Temperature=0 | | Temperature=0.5 | | Temperature=1 | |
|---|---|---|---|---|---|---|---|
| | | Acc. | Speedup | Acc. | Speedup | Acc. | Speedup |
| Vicuna | Vanilla | 0.29 | 1.00× | 0.25 | 1.00× | 0.21 | 1.00× |
| | **SpecBranch** | **0.29** | **1.95×** | **0.24** | **1.83×** | **0.22** | **1.75×** |
| LLaMA-3.1 | Vanilla | 0.93 | 1.00× | 0.92 | 1.00× | 0.90 | 1.00× |
| | **SpecBranch** | **0.93** | **3.67×** | **0.92** | **3.58×** | **0.90** | **3.54×** |

Table 6: SpecBranch achieves lossless acceleration across various model pairs and temperatures on GSM8K (Cobbe et al., 2021).

## E    EVALUATION DETAILS

For reproducibility, we discuss the experimental setup (Section 6) in detail and the source code of this project will be made available at a later time.

### E.1    DATA CONFIGURATIONS

In our experiments, we evaluate SpecBranch using the following dataset settings. The tasks include code generation, multilingual arithmetic reasoning, summarization, and Spec-Bench (a widely-adopted benchmark consisting of six sub-tasks) (Xia et al., 2024b). For code generation, we use HumanEval (Chen et al., 2021), a widely recognized benchmark comprising 164 problems. For arithmetic reasoning and multilingual inference, we use GSM8K (Cobbe et al., 2021), presenting their results side by side. Specifically, we sample the first 100 examples from GSM8K. For summarization, we use CNN/DM (Nallapati et al., 2016), and also sample the first 100 examples. The maximum generation length for these tasks is set to 512 tokens.

For Spec-Bench, we define distinct templates for each subtask. The template for Vicuna follows the official format, while for LLaMA and DeepSeek, the templates are as follows:

✧ **MT-Bench**: "A conversation between a curious user and an AI assistant, where the assistant provides helpful, detailed, and polite answers to the user's questions."

✧ **QA**: "A conversation between a curious user and an AI assistant, where the assistant gives helpful, detailed, and polite answers to the user's questions."

✧ **Summarization**: "Summarize: QUESTION TL;DR."

✧ **Translation**: "Translate German to English. German: QUESTION English."

✧ **Math**: "Let's think step by step."

✧ **RAG**: "A conversation between a curious user and an AI assistant, where the assistant gives helpful, detailed, and polite answers to the user's questions."

Specifically, due to the unique characteristics of LLaMA-3.1, we design a specialized template as follows: "You are a helpful, respectful, and honest assistant. Always answer as helpfully as possible, while being safe. Your answers should not include any harmful, unethical, racist, sexist, toxic, dangerous, or illegal content. Please ensure that your responses are socially unbiased and positive in nature. If a question does not make sense or is not factually coherent, explain why instead of providing an incorrect answer. If you don't know the answer, please do not share false information."

### E.2    MODEL CONFIGURATIONS

To validate performance, we select state-of-the-art open-source model pairs such as the LLaMA series (JackFram /LLaMA-68M, huggyLLaMA/LLaMA-7b), Vicuna (double7/vicuna-68M, lmsys/vicuna-13b-v1.3), Deepseek-Coder (deepseek-ai/deepseek-coder-1.3b-instruct, deepseek-ai/deepseek-coder-33b-instruct) and LLaMA-3.1 (meta-LLaMA/LLaMA-3.1-8B-Instruct, meta-LLaMA/LLaMA-3.1-70B-Instruct) for each task. All model weights are loaded in bfloat16 format for optimized GPU inference without quantization. As a draft model training-free method, SpecBranch does not modify any draft model parameters during evaluation. We summarize the model configuration in Table 7.

| Models | Layers | dim | FFN dim | Vocabulary size |
|--------|--------|-----|---------|-----------------|
| LLaMA 68M | 2 | 768 | 3072 | 32000 |
| LLaMA 7B | 32 | 4096 | 11008 | 32000 |
| Vicuna 68M | 2 | 768 | 3072 | 32000 |
| Vicuna 13B | 40 | 5120 | 13824 | 32000 |
| Deepseek 1.3B | 24 | 2048 | 5504 | 32256 |
| Deepseek 33B | 62 | 7168 | 19200 | 32256 |
| LLaMA-3.1 8B | 32 | 4096 | 14336 | 128256 |
| LLaMA-3.1 70B | 80 | 8192 | 28672 | 128256 |

Table 7: Model configurations.

### E.3 EVALUATION DETAILS

We report widely used metrics for speculative decoding (SD): Mean Accepted Length $M$, Wall-Time Speedup Ratio, and Speed (tokens/sec). Additionally, we introduce a new metric, Rollback Rate (RB), defined as $RB = \frac{\#\text{Rollback tokens}}{\#\text{Total tokens}}$, which quantifies computational waste due to invalid drafts. In SpecBranch, $M$ represents the continuously accepted length (Liu et al., 2024b), which is not the fixed length accepted in a single round of $\gamma$, but rather the higher accepted length achieved through multiple rounds of parallel generation, surpassing the performance of Vanilla SD. $RB$ specifically refers to the number of rollbacks during the draft model's forward times, excluding additional token loss due to branch and tree structures (since the impact of draft parallelism on acceleration is negligible).

All experiments, including the main results and ablation studies, are conducted on NVIDIA A100-PCIE-40G GPUs. Models with fewer than 8B parameters are run on a single device, 33B models on two devices, and 70B models on four devices. For inference, we set the batch size to 1 to match standard speculative decoding settings.

**Temperature Sampling** For SpecBranch and other baselines, we set the draft and target model temperature to 0 as greedy sampling. Since Top-$k$ resampling is adopted in SpecBranch, we derive the draft token confidence in temperature 1. More discussions about temperature sampling are detailed in Table 6.

**Resource Consumption** In the resource consumption experiments, we use the `NVIDIA Data Center GPU Manager (DCGM)` to monitor real-time GPU memory usage and power. Energy consumption is calculated by multiplying the average power by the total inference time over the entire benchmark.

All baselines, such as PEARL and SpS, are reproduced from their original papers and official codebases, using the optimal configurations reported by the authors. **Experiments are conducted under identical conditions to avoid introducing bias.** Importantly, for LLaMA-3.1 model pairs, the implementation of Lookahead requires transformers $= 4.36.2$, which conflicts with LLaMA 3.1's dependency on transformers $\geq 4.43.0$. A similar situation is also found in (Liu et al., 2024b).

### E.4 TRAINING DETAILS AND CROSS-TASK GENERALIZATION

**H-RAD Training** Our H-RAD training is performed offline on specific datasets, including our benchmarks. The training data for H-RAD pairs the feature vector $z_t$ from Eq. (4) with the corresponding three-class labels $s_t$. We implement a lightweight three-layer MLP with ReLU activation and dropout (rate$= 0.4$). The model architecture consists of two hidden layers (256 and 64 units) followed by a classification layer. Training is performed *offline* for 20 epochs with 32 batch size using the AdamW optimizer (Loshchilov & Hutter, 2017) (learning rate$= 5 \times 10^{-5}$, weight decay$= 1 \times 10^{-4}$). To address class imbalance issues, we employ SMOTE (Synthetic Minority Oversampling Technique) data augmentation (Chawla et al., 2002), specifically targeting underrepresented classes in H-RAD. The augmentation process involves standardizing features, applying SMOTE with $k = 5$ nearest neighbors, and then inverse transforming the synthetic samples. Additionally, we utilize label smoothing (smoothing$= 0.1$) in the loss function to prevent overconfident predictions and improve generalization. The training process incorporates several optimization strategies:

✧ Learning rate scheduling with ReduceLROnPlateau (factor= 0.5, patience= 2).

✧ Early stopping with 5 epochs patience.

✧ Gradient clipping with max norm of 1.0.

✧ Weighted random sampling to balance class distributions.

The training converges within 5 minutes on a single NVIDIA A100 GPU and avoids the need for costly online fine-tuning while maintaining adaptability to diverse domains. The model achieves balanced performance across the three classes, as validated through confusion matrix analysis and t-SNE visualization of the learned feature space.

**Cross-Task Generalization**     First, we emphasize that H-RAD consists of a lightweight MLP predictor, which involves very low offline training costs. Re-training and deployment on new datasets require minimal additional expenses. To better demonstrate its generalization capacity across different datasets and tasks, we conduct more experiments and analyses.

We define H-RAD pre-trained with Spec-Bench as the experimental module, referred to as SpecBranch (Spec). As shown in Table 3, our method performs exceptionally well across six different sub-tasks of Spec-Bench, highlighting H-RAD's strong cross-task performance.

Additionally, we transfer pre-trained H-RAD (Spec) directly to HumanEval, GSM8K, and CNN/DM and compare with the original SpecBranch trained on specific datasets. As shown in Table 8, we see a little performance drop of less than 10%, with an average decrease of only **5%**, yet it still outperforms the baseline PEARL, showcasing H-RAD's generalization and robustness.

| Models | Methods | HumanEval | | GSM8K | | CNN/DM | | Speed | Avg. |
|---|---|---|---|---|---|---|---|---|---|
| | | M | Speedup | M | Speedup | M | Speedup | (tokens/s) | Speedup |
| Vicuna | PEARL | 3.11 | 2.02× | 2.83 | 1.61× | 2.89 | 1.68× | 53.31 | 1.77× |
| | SpecBranch | 3.69 | 2.47× | 3.29 | 1.95× | 3.21 | 1.89× | 62.57 | 2.10× |
| | **SpecBranch(Spec)** | 3.45 | 2.31×(6.48%) | 3.13 | 1.78×(8.72%) | 3.15 | 1.81×(4.24%) | 58.81 | 1.97×(6.20%) |
| LLaMA-3.1 | PEARL | 17.28 | 3.75× | 14.33 | 3.35× | 7.51 | 3.04× | 24.03 | 3.38× |
| | SpecBranch | 21.74 | 4.02× | 18.08 | 3.67× | 9.41 | 3.37× | 26.27 | 3.69× |
| | **SpecBranch(Spec)** | 20.53 | 3.93×(2.31%) | 17.58 | 3.48×(5.18%) | 9.15 | 3.19×(5.35%) | 25.03 | 3.52×(4.71%) |

Table 8: Cross-Task Generalization of SpecBranch (H-RAD pre-trained with Spec-Bench).

This generalization capability can be partially attributed to the hybrid prediction of H-RAD, where intermediate token results are determined by confidence, rather than relying on dataset distributions. Thus, H-RAD achieves better generalization than a purely explicit method. Furthermore, the features of the target model provide guidance for token acceptance, which is influenced more by the model pair. Across different tasks, the predictive effectiveness of these features remains highly consistent, demonstrating that H-RAD exhibits **strong generalization across target tasks or domains** for specific model pairs. Online training of H-RAD is a promising future direction since H-RAD can learn in real-time from historical draft lengths and target model features, thereby enhancing its generalization capacity in real-world deployments.

# F  MORE EXPERIMENTAL RESULTS

## F.1  MORE COMPREHENSIVE COMPARISONS WITH THE TREE-BASED METHODS

To further elucidate the differences between our method and tree-based methods, we conduct additional comparisons as described in the following.

**Compared to training-required tree-based methods:**

Since training-required methods such as EAGLE (Li et al., 2024a) incur substantial training costs, it is unfair to directly compare with EAGLE since our method is model training-free and such cost is well-recognized in the field. To further highlight the efficiency of SpecBranch, we provide a detailed breakdown of the draft model and predictor training costs (refer to Table **??** for additional details).

We compare the draft model and predictor training costs of SpecBranch with three representative training-required methods: EAGLE, Medusa (Cai et al., 2024) and Kangaroo (Liu et al., 2024a). The

| Method | Training Cost Details |
|---|---|
| **EAGLE** | **Model:** 1-2 days (LLaMA-2-13B, 8 RTX 3090 GPUs) 
 **Predictor:** None |
| **Medusa** | **Model:** 5 hours (Vicuna 7B, 1 NVIDIA A100-PCIE GPU) 
 **Predictor:** None |
| **Kangaroo** | **Model:** 24 hours (Vicuna-7B, 8 NVIDIA V100-PCIE GPUs) 
 **Predictor:** None |
| **SpecBranch** | **Model:** None 
 **Predictor:** $\leq$ 5 mins (1 NVIDIA A100-PCIE GPU) |

Table 9: Comparison of draft model and predictor training costs.

latter two also require time-intensive fine-tuning on large datasets. In contrast, the draft model of SpecBranch does not require training. H-RAD only involves training a lightweight MLP predictor that takes a few minutes on a single NVIDIA A100 GPU and we can see the training cost is **250×** lower than EAGLE.

To further highlight SpecBranch's exceptional performance in terms of latency, we provide key metrics including the mean accepted tokens (M) and speedup, for comparison between SpecBranch and the EAGLE family on LLaMA-2 Chat 7B&70B.

| Models | Methods | HumanEval | | GSM8K | | CNN/DM | | Speed | Avg. |
|---|---|---|---|---|---|---|---|---|---|
| | | M | Speedup | M | Speedup | M | Speedup | (tokens/s) | Speedup |
| LLaMA-2 | EAGLE | 4.45 | 3.51× | 3.97 | 3.09× | 3.78 | 2.98× | 22.35 | 3.20× |
| | EAGLE-2 | 5.46 | **3.78×** | 4.49 | **3.52×** | 4.98 | **3.48×** | **25.13** | **3.59×** |
| | PEARL | 15.34 | 3.21× | 13.29 | 3.07× | 12.98 | 2.96× | 21.56 | 3.08× |
| | **SpecBranch** | **18.27** | 3.59× | **15.98** | 3.37× | **14.02** | 3.23× | 23.73 | 3.39× |

Table 10: Comparison with training-required tree methods on LLaMA-2 7B&70B.

As shown in the Table 10, SpecBranch demonstrates an impressive speedup of 3.59×, surpassing EAGLE's 3.51×. While it is slightly behind EAGLE-2 (Li et al., 2024b) 's 3.78×, the performance closeness already indicates the significant potential of SpecBranch's branch-parallel framework and the H-RAD predictor.

Additionally, H-RAD is an independent component that can be "plug and play" into EAGLE. While EAGLE-2 dynamically adjusts the tree width through token confidence, H-RAD further optimizes the tree depth. We also point out that the parallel paradigm can be combined with EAGLE. For example, we can train a 4-layer EAGLE model with early-exit capability, enabling dynamic adjustment of the draft model's layers based on context, using parameters from {1, 2, 3, 4} layers. By combining this early-exit, multi-layer draft model with the target model for parallel prediction, we can form a new branching parallel framework.

| Models | Methods | HumanEval | | GSM8K | | CNN/DM | | Speed | Avg. |
|---|---|---|---|---|---|---|---|---|---|
| | | M | Speedup | M | Speedup | M | Speedup | (tokens/s) | Speedup |
| LLaMA-2 | REST | 1.97 | 1.78× | 1.65 | 1.46× | 1.68 | 1.57× | 10.99 | 2.28× |
| | Ouroboros | 5.16 | 2.10× | 5.96 | 2.58× | 3.28 | 1.57× | 14.60 | 2.08× |
| | SWIFT | 4.27 | 1.56× | 2.99 | 1.43× | 3.87 | 1.45× | 10.15 | 1.48× |
| | PEARL | 15.34 | 3.21× | 13.29 | 3.07× | 12.98 | 2.96× | 21.56 | 3.08× |
| | **SpecBranch** | **18.27** | **3.59×** | **15.98** | **3.37×** | **14.02** | **3.23×** | **25.83** | **3.39×** |

Table 11: Comparison to model training-free tree methods on LLaMA-2 7B&70B.

**Compared to model training-free tree methods:**

To justify SpecBranch against tree-structure methods, we have also conducted more experiments of the recent model training-free methods, such as REST (He et al., 2023) (retrieval tree structure

self-SD), Ouroboros (Zhao et al., 2024) (lookahead tree structure), and SWIFT (Xia et al., 2024a) (layer-skip tree structure self-SD), evaluated on Spec-Bench using LLaMA-2 Chat 7B&70B.

The comparisons in the Table 11 show that SpecBranch outperforms other tree-structure methods. Moreover, the parallel framework can be orthogonally combined with these methods, offering significant future potential.

## F.2 EVALUATION RESULTS OF VARYING $k_{max}$

The number of branches $k$ is a crucial hyperparameter, which is discussed in Section 6.2 Resource Consumption. Fig. 7 demonstrates that memory consumption for parallel branches increases only slightly with $k$ ranging from 1 to 6. Furthermore, we perform more discussion and new experiments to report how varying $k_{max}$ affects overall speedup and rollback rate (RB). As shown in Table 12, we conduct SpecBranch experiments on HumanEval for Vicuna 68M&13B and LLaMA-3.1 8B&70B under varying $k_{max}$.

| Model | $k_{\mathbf{max}} = 1$ | | $k_{\mathbf{max}} = 2$ | | $k_{\mathbf{max}} = 4$ | | $k_{\mathbf{max}} = 6$ | | $k_{\mathbf{max}} = 12$ | | $k_{\mathbf{max}} = 18$ | |
|---|---|---|---|---|---|---|---|---|---|---|---|---|
| | RB | Speedup | RB | Speedup | RB | Speedup | RB | Speedup | RB | Speedup | RB | Speedup |
| Vicuna 68M&13B | 87.56% | 2.05× | 62.13% | 2.16× | 48.02% | 2.28× | 45.14% | 2.36× | **39.60%** | **2.47×** | 38.92% | 2.48× |
| LLaMA-3.1 8B&70B | 19.63% | 3.76× | 15.63% | 3.88× | 11.63% | 3.97× | **9.51%** | **4.02×** | 8.93% | 4.05× | 8.87% | 4.06× |

Table 12: Performance comparison under different $k_{\mathrm{max}}$ values.

We observe that as $k_{max}$ increases, SpecBranch initially exhibits rapid acceleration, then slows down as speedup diminishes. Conversely, the rollback rate behaves in an opposite manner. When $k_{max} = 1$, there is only one parallel branch. As the number of branches increases, more candidate branching points are available, improving the acceptance rate of unconfident tokens. However, this comes at the cost of higher memory consumption.

(1) For poorly aligned model pairs, the lower token acceptance rate necessitates more branches to increase the acceptance of unconfident tokens, thereby reducing rollback and improving parallel efficiency to achieve a trade-off. Accordingly, a lightweight draft model can handle more memory, enabling more branches to be processed in parallel.

(2) For well-aligned model pairs, the improved alignment between the target and draft models mitigates the impact of rollback, meaning that a smaller $k_{max}$ is sufficient to achieve the desired trade-off.

## F.3 LATENCY STABILITY ACROSS SEQUENCES

In principle, for SpecBranch or any other SD methods, latency is highly correlated with acceptance. We define the acceptance for each iteration as the number of accepted tokens in that round divided by $\gamma$ (draft length one iteration). In practical scenarios, the acceptance for each iteration varies significantly, as shown in the Appendix F.9. This is why finding an optimal draft length is challenging and only dynamic draft lengths can effectively enable acceleration.

However, when we shift the time scale from iterations to requests, we find that the average acceptance rate for each request is quite consistent, with $\alpha$ being more dependent on model alignment. As shown in the Table 13 (HumanEval with LLaMA 68M&7B), when we treat a request as the unit of measurement, the latency of SpecBranch remains quite stable.

| Method | Response Time per Request (s) | | | | | | | | | | Std. Dev. |
|---|---|---|---|---|---|---|---|---|---|---|---|
| | 1 | 2 | 3 | 4 | 5 | 6 | 7 | 8 | 9 | 10 | (n=100) |
| PEARL | 4.55 | 4.80 | 5.21 | 4.92 | 4.74 | 5.90 | 5.58 | 4.98 | 5.77 | 6.04 | 0.589 |
| **SpecBranch** | 4.10 | 4.31 | 4.67 | 4.39 | 4.28 | 5.37 | 5.00 | 4.44 | 5.10 | 5.02 | 0.423 |

Table 13: Response time and standard deviation for different methods across requests

Compared to PEARL, by using H-RAD to efficiently predict unconfident tokens, we reduce token rollback, leading to better stability in latency. Specifically, the standard deviation for 100 requests is only 0.423, much lower than 0.589 of PEARL.

## F.4 TIME AND ENERGY CONSUMPTION

We have provided some evaluation results of time and energy consumption in Section 6.2. In this section, we conduct more experiments on the time consumption of various model pairs on HumanEval while we further test the energy consumption of various model pairs on HumanEval and GSM8K.

**Time consumption** As shown in Table 14, regardless of model size, the time spent on H-RAD prediction and communication between multiple GPUs is almost negligible compared to the total inference time for a single step. This indicates that the H-RAD module retains its lightweight nature with minimum resource consumption and the inter-GPU communications have low operational overhead as well. On the other hand, the time span for the draft stage and the target model verification stage is nearly equal. This is consistent with our prior results that SpecBranch effectively implements parallelism between these two stages to alleviate the mutual waiting bubbles.

| Modules | LLaMA 68M&7B | Vicuna 68M&13B | Deepseek 1.3&33B | LLaMA-3.1 8B&70B |
|---|---|---|---|---|
| **H-RAD Predict** | **0.26 ms** | **0.27 ms** | **0.31 ms** | **0.28 ms** |
| Communication | 0.21 ms | 0.31 ms | 0.26 ms | 0.26 ms |
| Draft Stage | 20.8 ms | 30.9 ms | 58.3 ms | 125.1 ms |
| Verification Stage | 21.7 ms | 31.4 ms | 56.1 ms | 128.0 ms |

Table 14: Time cost of each module (per step) on the HumanEval dataset.

**Energy consumption** We use the `NVIDIA Data Center GPU Manager (DCGM)` toolkit to monitor the real-time power consumption and collect relevant traces. Energy consumption is calculated by multiplying the average power by the total inference time over the entire benchmark. The results are provided in Table 15 and Table 16.

For poorly aligned models (LLaMA, Vicuna), SpecBranch with its rollback-aware dynamic draft length through H-RAD, significantly reduces redundant tokens, thereby lowering energy consumption compared to PEARL and SpS. Unlike PEARL, which pre-verifies only the first token using the target model in parallel, SpecBranch utilizes H-RAD to predict all tokens during the draft stage. This approach reduces the number of forward passes required by the target model. For LLaMA 68M&7B on HumanEval, SpecBranch reduces the target model forward passes to **32,285**, compared to **44,949** in PEARL, resulting in lower energy consumption for target model inference. For better-aligned models (Deepseek, LLaMA-3.1), where token rollback is significantly reduced, parallel efficiency improves considerably. While energy savings are less pronounced, SpecBranch still outperforms PEARL.

Theoretically, Parallel SD incurs negligible energy consumption compared to Vanilla SD. The speedup achieved through parallelism offsets the additional power consumption caused by extra forward passes in the target model. However, the parallelism and communication overhead still contribute to some energy consumption in real-world deployments. Overall, SpecBranch, with its hybrid rollback-aware draft structure, significantly optimizes energy consumption compared to PEARL. Notably, for poorly aligned models, it outperforms SD in terms of energy efficiency.

| Methods | LLaMA 68M&7B | Vicuna 68M&13B | Deepseek 1.3&33B | LLaMA-3.1 8B&70B |
|---|---|---|---|---|
| SpS | 217 KJ | 383 KJ | 565 KJ | 1021KJ |
| PEARL | 287 KJ | 445 KJ | 631 KJ | 1231 KJ |
| **SpecBranch** | **156 KJ** | **251 KJ** | **582 KJ** | **1092 KJ** |

Table 15: Energy cost of SpecBranch and baseline methods on the HumanEval dataset.

## F.5 MORE EVALUATION RESULTS ON THE SPEC-BENCH BENCHMARK

As illustrated in Section 6, we provide more evaluation results of SpecBranch in Table F.5 with both LLaMA 68M&7B and Deepseek 1.3&33B on Spec-Bench. Notably, LLaMA 3.1 is a more

| Methods | LLaMA 68M&7B | Vicuna 68M&13B | Deepseek 1.3&33B | LLaMA-3.1 8B&70B |
|---------|--------------|----------------|------------------|-------------------|
| SpS | 146 KJ | 250 KJ | 393 KJ | 741 KJ |
| PEARL | 193.6 KJ | 295 KJ | 428 KJ | 901 KJ |
| **SpecBranch** | **128 KJ** | **178 KJ** | **395 KJ** | **791 KJ** |

Table 16: Energy cost of SpecBranch and baseline methods on the GSM8K dataset.

recent LLM version that requires the transformer version to be greater than $4.43.0$. Due to this incompatibility, we cannot reproduce the results of baseline Lookahead Decoding as described before.

| Models | Methods | MT Bench | | QA | | Sum | | Math | | RAG | | Trans | | Avg. |
|--------|---------|------|---------|------|---------|------|---------|------|---------|------|---------|------|---------|------|
| | | M | Speedup | M | Speedup | M | Speedup | M | Speedup | M | Speedup | M | Speedup | |
| LLaMA | SpS | 2.78 | 1.77× | 4.43 | 2.57× | 2.62 | 1.51× | 5.02 | 2.50× | 2.39 | 1.28× | 5.28 | 2.90× | 2.09× |
| | AdaEDL | 2.74 | 1.86× | 4.25 | 2.59× | 2.56 | 1.54× | 4.83 | 2.43× | 2.27 | 1.39× | 4.93 | 2.96× | 2.13× |
| | Lookahead | 1.58 | 1.31× | 1.62 | 1.34× | 1.59 | 1.32× | 1.73 | 1.51× | 1.31 | 1.15× | 1.36 | 1.23× | 1.31× |
| | PEARL | 2.83 | 2.14× | 5.98 | 3.15× | 2.79 | **1.85×** | 7.58 | 3.37× | 2.48 | 1.43× | 7.65 | 4.03× | 2.66× |
| | **SpecBranch** | **3.01** | **2.34×** | **6.57** | **3.53×** | **2.87** | 1.76× | **8.57** | **3.77×** | **3.06** | **1.64×** | **8.57** | **4.32×** | **2.89×** |
| Vicuna | SpS | 2.63 | 1.74× | 2.47 | 1.55× | 2.58 | 1.70× | 2.37 | 1.55× | 2.47 | 1.56× | 2.57 | 1.65× | 1.64× |
| | AdaEDL | 2.50 | 1.80× | 2.43 | 1.67× | 2.64 | 1.72× | 2.31 | 1.62× | 2.21 | 1.57× | 2.45 | 1.75× | 1.69× |
| | Lookahead | 1.56 | 1.31× | 1.41 | 1.23× | 1.49 | 1.25× | 1.71 | 1.46× | 1.38 | 1.15× | 1.32 | 1.10× | 1.25× |
| | PEARL | 2.62 | 1.78× | 2.45 | 1.64× | **2.78** | **1.83×** | 2.63 | 1.67× | 2.61 | 1.66× | 2.89 | 2.05× | 1.77× |
| | **SpecBranch** | **3.11** | **2.09×** | **2.67** | **1.83×** | 2.72 | 1.78× | **2.86** | **1.89×** | **2.83** | **1.86×** | **3.32** | **2.30×** | **1.96×** |
| Deepseek | SpS | 3.99 | 2.03× | 4.02 | 2.12× | 4.08 | 2.02× | 3.93 | 2.06× | 3.95 | 1.94× | 4.37 | 2.21× | 2.06× |
| | AdaEDL | 3.54 | 2.26× | 3.81 | 2.33× | 3.74 | 2.03× | 3.81 | 2.28× | 3.69 | 2.03× | 4.12 | 2.38× | 2.22× |
| | Lookahead | 1.78 | 1.51× | 1.81 | 1.64× | 1.59 | 1.42× | 1.75 | 1.53× | 1.53 | 1.32× | 1.74 | 1.50× | 1.49× |
| | PEARL | 6.60 | 2.77× | 5.15 | **2.91×** | 5.81 | 2.68× | 7.14 | 2.79× | 5.77 | 2.65× | 6.20 | 3.09× | 2.81× |
| | **SpecBranch** | **8.31** | **3.02×** | **5.71** | 2.87× | **6.67** | **2.95×** | **8.92** | **3.21×** | **6.82** | **2.76×** | **6.86** | **3.28×** | **3.02×** |
| LLaMA-3.1 | SpS | 4.67 | 2.31× | 4.57 | 2.27× | 5.09 | 1.98× | 5.01 | 2.44× | 5.08 | 2.02× | 5.52 | 2.57× | 2.27× |
| | AdaEDL | 4.31 | 2.43× | 4.23 | 2.30× | 4.83 | 2.05× | 4.94 | 2.46× | 4.86 | 2.13× | 5.24 | 2.65× | 2.34× |
| | Lookahead | - | - | - | - | - | - | - | - | - | - | - | - | - |
| | PEARL | 8.46 | 2.96× | 8.37 | 3.27× | 9.10 | 3.32× | 12.53 | 3.39× | 8.35 | **3.41×** | 12.59 | 4.22× | 3.43× |
| | **SpecBranch** | **10.85** | **3.24×** | **10.59** | **3.45×** | **11.40** | **3.63×** | **15.76** | **3.78×** | **9.16** | 3.40× | **16.64** | **4.51×** | **3.67×** |

Table 17: More evaluation results on Spec-Bench. "–" indicate incompatibility: baseline implementations (transformers $= 4.36.2$) and LLaMA 3.1's dependency on $\geq 4.43.0$.

We draw several interesting findings from Table F.5: 1) SpecBranch demonstrates consistent speedups across six subtasks, regardless of the model alignment quality; 2) In particular, the LLaMA 68M&7B model combination performs exceptionally well, due to the capacity of the LLaMA model, which excels in the Math, QA, and Translation tasks, with an average accepted length even surpassing DeepSeek. However, there are significant performance variations across other tasks. This suggests that the alignment between the small draft models and the large target model may vary across tasks, leading to highly task-specific effects. These findings also indicate incorporate multiple draft models for speculative sampling in future research. Different draft models can be tailored to specific tasks under the Mixture-of-Experts (MoE) principles to further enhance inference efficiency.

## F.6    MORE EVALUATION RESULTS OF ROLLBACK RATIO

As discussed in Section 6, rollback is an important metric for the parallel efficiency. Here, we provide more evaluation results on HumanEval, GSM8K, CNN/DM, and Spec-Bench, as shown in Fig. 11. The results demonstrate that SpecBranch achieves significantly lower rollback ratios across various datasets and subtasks compared to other methods. This is consistent with our prior justification that the H-RAD module plays an essential role in mitigating rollback – the additional evaluations in Fig. 11 further validate the generalization capacity of H-RAD in different tasks. It is worth mentioning that for poorly aligned model combinations (e.g., Vicuna, LLaMA), SpecBranch reduces the rollback ratio by nearly 50% compared to PEARL. Even for better-aligned model pairs (e.g., Deepseek, LLaMA-3.1), it achieves about 10% reduction. These results indicate the potential of SpecBranch in resource-constrained environments where the draft model sizes are typically restrained. The proposed framework can reduce the percentage of rollbacks as well as save computation/energy resources in the long run.

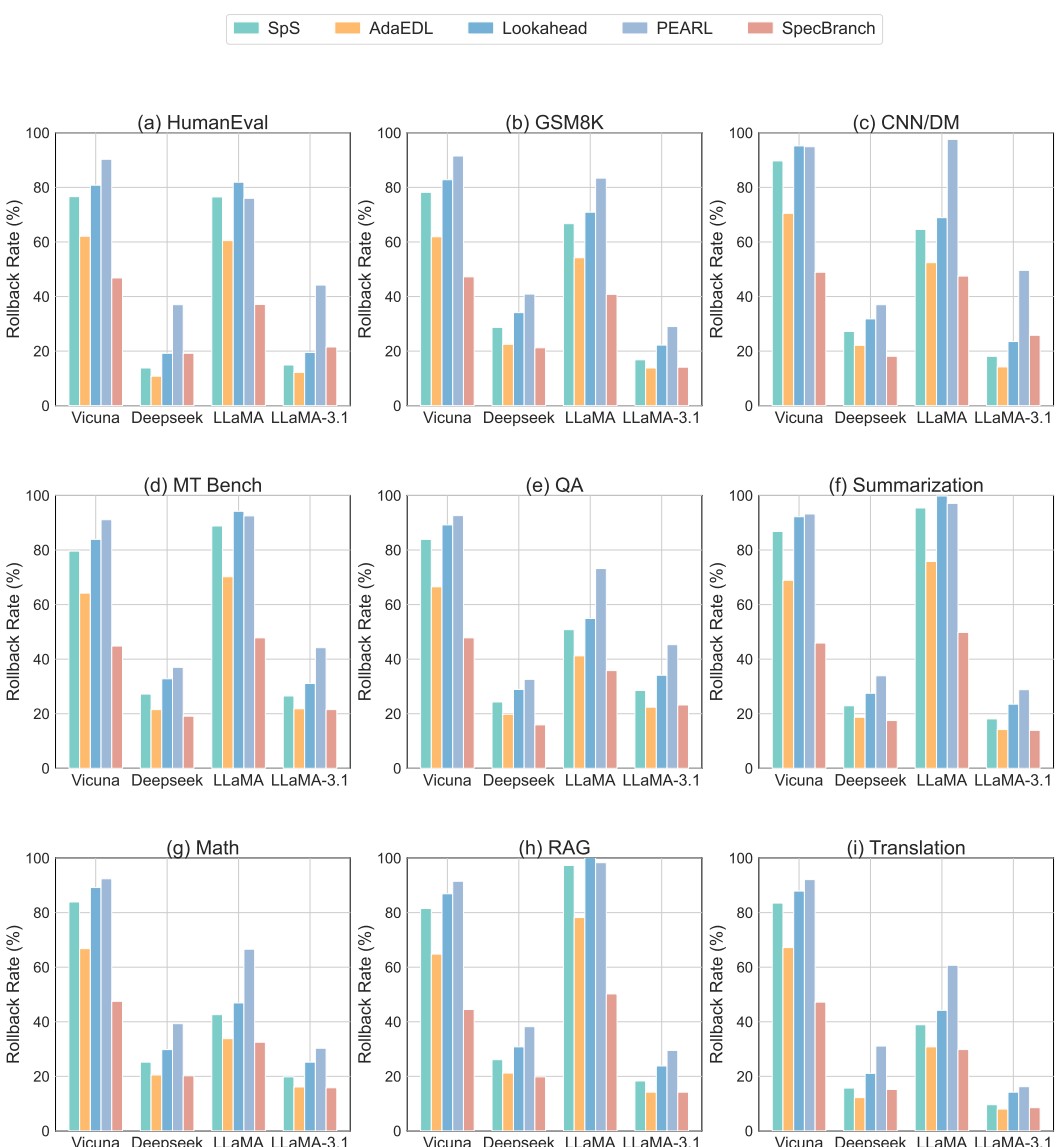

Figure 11: Comparison of Rollback Rates on HumanEval, GSM8K, CNN/DM, Spec-Bench for different model combinations.

## F.7 MORE RESULTS OF TOKEN DISTRIBUTION

Recall that in Section 4.1 and Fig. 1(b), we introduce a truncated geometric distribution (Leviathan et al., 2023)(shown in Fig. 1(b),

$$P(X = k) = (1 - \alpha) \cdot \alpha^k \cdot \mathbb{I}(k < \gamma) + \alpha^\gamma \cdot \mathbb{I}(k = \gamma), \tag{12}$$

where $\alpha^\gamma$ is the probability of *full acceptance* and $1 - \alpha^\gamma$ is the probability of *rollback*. To validate this, we present additional token distribution results for Vicuna 68M&13B and Deepseek 1.3B&33B on HumanEval and GSM8K in Figs. 12 and 13. These results show that the token acceptance distribution closely follows the truncated geometric distribution, which is consistent with our prior statements of a bimodal phenomenon extracted from the target model features. This has laid the foundation for the H-RAD module to perform token length predictions with high fidelity.

In particular, Fig. 12 shows that for poorly aligned models, the acceptance rate $\alpha$ is low, and the truncated geometric distribution peaks at All-Reject. In this scenario, the rollback in SD is exacerbated, particularly in Parallel SD, since it only achieves parallelism under the All-Accept condition. Given the low proportion of All-Accept in this token distribution, the efficiency of parallelism is significantly

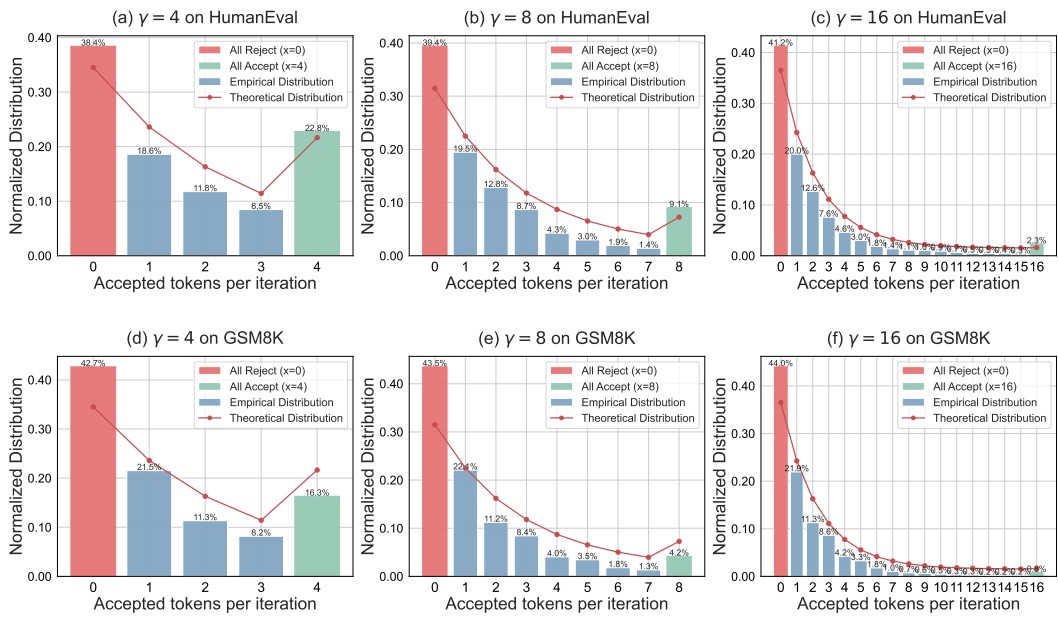

Figure 12: More evaluation results demonstrate that the distribution of accepted tokens generally follows a truncated geometric distribution of different token length $\gamma$ of Vicuna 68M&13B.

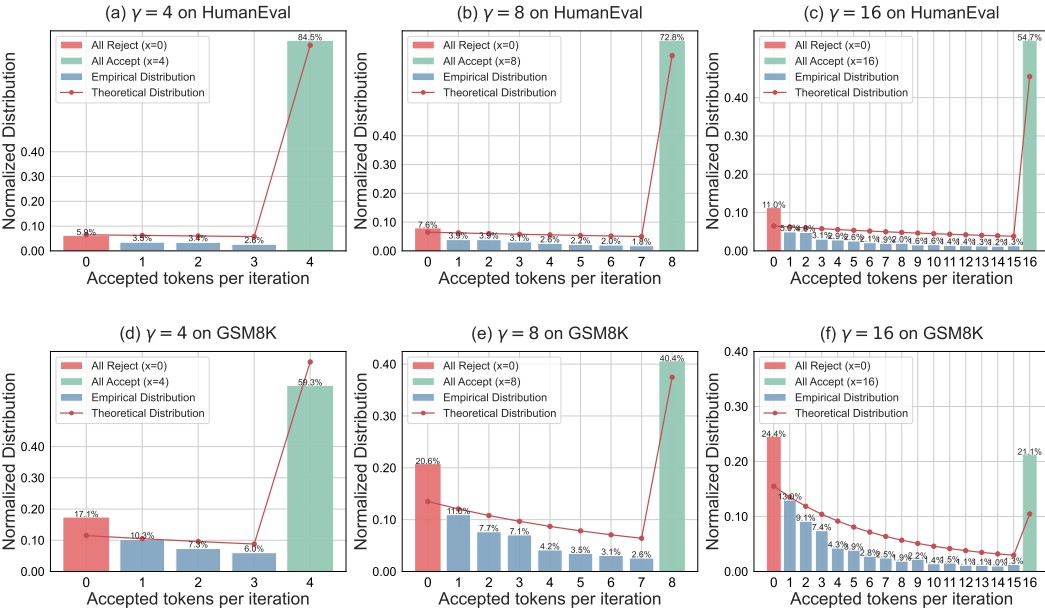

Figure 13: Distribution of accepted tokens generally follows a truncated geometric distribution of different token length $\gamma$ of Deepseek 1.3B &33B.

impacted by rollback. This highlights our motivation to jointly consider rollback with parallelism for traditional SD.

For better-aligned models, Fig. 13 shows that the acceptance rate $\alpha$ is high and the truncated geometric distribution peaks at All-Accept. In this scenario, the rollback effects in SD are undermined, with the balance between rollback and parallelism leaning towards parallelism. As a result, the parallel framework shows significant acceleration compared to vanilla SD, even approaching the theoretical speedup limit with well-aligned models. However, it is evident that as the draft length increases, the impact of rollback becomes non-negligible. SpecBranch effectively balances the trade-off between these factors.

## F.8 EVALUATION RESULTS OF IMPLICIT DISTRIBUTION

As discussed in Sections 4.2 and 6.2, we primarily analyze the sensitivity of hyperparameters in the implicit methods. Here, we further examine the top-1 implicit distribution under different experimental setups. We conduct extensive experiments with LLaMA 68M&7B, Deepseek 1.3B&33B, and LLaMA-3.1 8B&70B on HumanEval, GSM8K, and CNN/DM under various temperature settings.

**Task Sensitivity**    We first explore the task sensitivity of two main implicit values: *confidence* and *entropy*, which measure confidence by $\max_{x_i} q(x_i)$ (Du et al., 2024) and entropy as $1 - \sqrt{\lambda H(x_i)}$ (Agrawal et al., 2024) against pre-determined thresholds $\epsilon$. Fig. 14 illustrates that the implicit values have different distributions across tasks. In summarization tasks (CNN/DM), both the average accepted confidence ($0.91$) and entropy ($0.77$) are significantly higher than in other tasks. Meanwhile, the rejected implicit values also vary notably ($0.26$ to $0.45$), especially for entropy in summarization tasks. This indicates that the implicit distribution is highly task-sensitive, making the selection of the stop threshold $\epsilon$ static and finding an optimal value difficult.

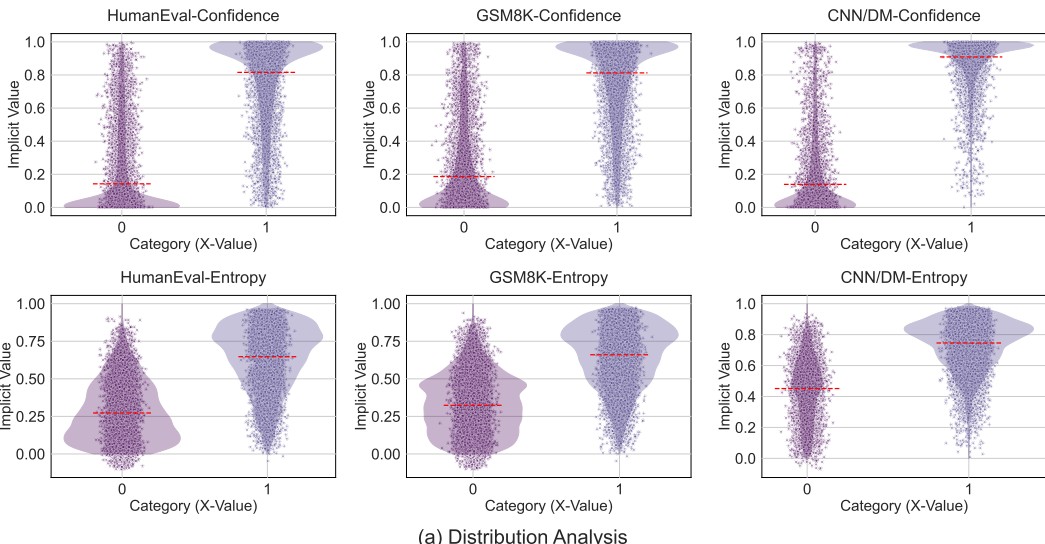

(a) Distribution Analysis

Figure 14: The top-1 implicit values (Confidence) distribution of LLaMA 68M&7B on HumanEval, GSM8k and CNN/DM with temperature $= 1$. Category 0 denotes the top-1 values of rejected draft tokens while 1 denotes the corresponding values of accepted tokens.

On the other hand, we observe that compared to entropy, confidence has a clearer and more distinct distribution for accepted and rejected tokens. This is why confidence is chosen for H-RAD, as it provides higher fidelity. Additionally, we note that around $0.5$, both confidence and entropy show considerable overlap between accepted and rejected values, which indicates a key limitation of the implicit methods.

**Model Sensitivity**    The distribution of entropy is less effective than that of confidence, so we further test the model sensitivity of implicit methods (confidence) on three different model sizes: LLaMA 68M&7B, Deepseek 1.3B&33B, and LLaMA-3.1 8B&70B. As shown in Fig. 15, the average accepted confidence of better-aligned models ($0.98$) is higher than that of poorly aligned models ($0.79$), and the same holds for rejected confidence ($0.27$ against $0.11$). This demonstrates that the confidence distribution is sensitive to model pairs, with lower average confidence in poorly aligned models, resulting in a higher rate of rollback.

**Temperature Sensitivity**    Temperature plays an important role in the draft model's sampling process. Thus, we conduct additional analysis on the temperature parameter. As shown in Fig. 16, we observe a sharp variation in the confidence distribution with temperature, especially for rejected tokens. When temperature $= 1$, the draft model generates tokens with higher randomness, leading to a more distinct and separated confidence distribution. At temperature $= 0.7$, the average rejected confidence rises to $0.55$, overlapping more with accepted confidence. When temperature $= 0.2$, the randomness of the draft model's sampling decreases, causing the rejected and accepted confidence distributions to overlap, making it difficult to distinguish early stop tokens. From the above, we

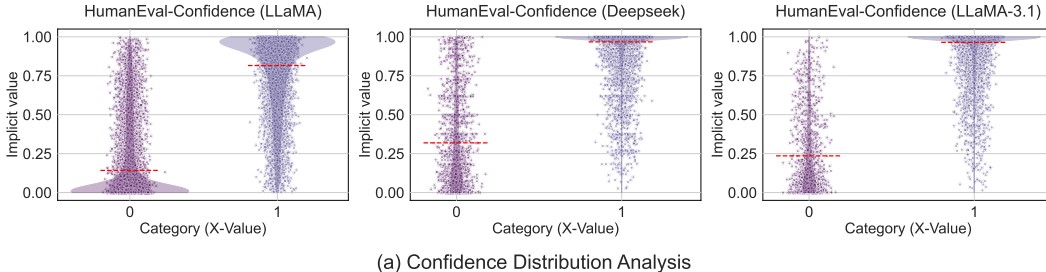

(a) Confidence Distribution Analysis

Figure 15: The top-1 implicit values (Confidence) distribution of LLaMA 68M&7B, Deepseek 1.3B&33B, and LLaMA-3,1 8B&70B on HumanEval with temperature = 1. Category 0 denotes the top-1 values of rejected draft tokens, while 1 denotes the corresponding values of accepted tokens.

conclude that higher temperatures increase randomness and allow the confidence distribution to better reflect whether draft tokens are accepted by the target model or not. Notably, SpecBranch uses temperature = 1 for top-$k$ sampling and confidence selection, optimizing the use of implicit values for better representation.

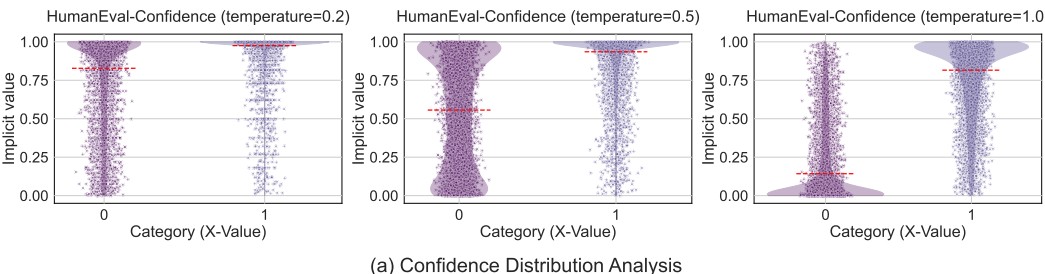

(a) Confidence Distribution Analysis

Figure 16: The top-1 implicit values (Confidence, Entropy) distribution of LLaMA 68M&7B on HumanEval with temperature = 0.2, 0.5 and 1.

Based on the above discussion, we summarize the advantages of H-RAD over implicit methods. H-RAD significantly reduces the frequency of implicit confidence calls at the token-level by leveraging explicit target model features. This mitigates error accumulation in implicit confidence, desensitizes and reduces the dependency on thresholds. Moreover, H-RAD improves the accuracy of dynamic draft structures, resulting in a significant reduction of rollback tokens. Specifically, our H-RAD predictor is invoked only once during each draft process, incurring far lower overhead compared to the token-level implicit training predictors (Huang et al., 2024).

### F.9 THE OPTIMAL DRAFT LENGTH

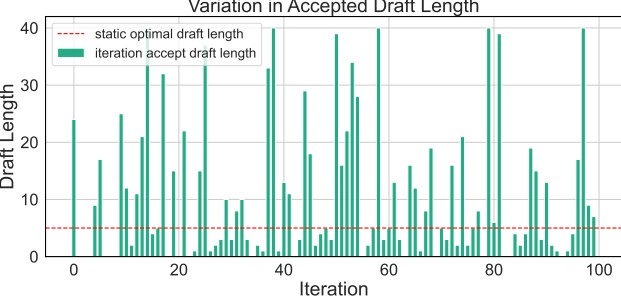

Figure 17: Variation of the optimal draft length over different iterations.

In Section 4.1, we have discussed the theoretical solution under rollback. However, this theoretical analysis only reflects the statistical properties rather than the runtime dynamics. As shown in Fig. 17, the actual optimal accepted draft length is context-dependent and varies significantly across different iterations and the same phenomenon is observed in (Zhang et al., 2024; Liu et al., 2024b). Hence,

fixed draft length would result in substantial rollback and token waste. This highlights the need for adaptive control of $\gamma$ rather than relying on static configurations.

### F.10 MORE ABLATION STUDY OF THE ORIGINAL MODULE AND THE HYPERPARAMETER SENSITIVITY

**Component Analysis**   To better demonstrate the effect of branch resampling by comparing with the original module, we conduct more experiments comparing vanilla SD and SpecBranch without H-RAD on the Spec-Bench of LLaMA 3.1 8B&70B. As shown in Table 18, SpecBranch shows a significant speedup ($3.36\times$) over SD ($2.27\times$).

| Methods | MTBench | | QA | | Sum | | Math | | RAG | | Trans | | Avg. | |
|---|---|---|---|---|---|---|---|---|---|---|---|---|---|---|
| | M | Speedup | M | Speedup | M | Speedup | M | Speedup | M | Speedup | M | Speedup | M | Speedup |
| **Vanilla SD** | 4.67 | 2.31× | 4.57 | 2.27× | 5.09 | 1.98× | 5.01 | 2.44× | 5.08 | 2.02× | 5.52 | 2.57× | 4.99 | 2.27× |
| **SpecBranch w/o H-RAD** | **8.14** | **2.91×** | **8.21** | **3.17×** | **8.87** | **3.37×** | **11.49** | **3.35×** | **7.98** | **3.24×** | **11.73** | **4.15×** | **9.40** | **3.36×** |

Table 18: Comparing the original module of the vanilla SD and SpecBranch without H-RAD.

**Hyperparameter sensitivity**   To improve alignment with established practices, we conduct more experiments with the Branch-Parallel structure of LLaMA 68M&7B on HumanEval and the explicit method. The results shown in Table 19 confirm H-RAD's high acceleration efficiency and lower sensitivity to hyperparameters.

**More Discussion**   We further clarify connections among SpecBranch's components: 1) Branch-Parallel is a holistic structure that covers draft parallelism as well as branch resampling. Single-sequence parallelism (e.g., PEARL (Liu et al., 2024b)) is a special case with $k_{max} = 1$. Detailed results on SpecBranch's performance across $k_{max}$ values are shown in Appendix F.2. The branch-parallel framework uses top-$k$ sampling for unconfident tokens (vs. top-1 in single-sequence parallelism) to mitigate potential token rollbacks. 2) Isolating branches from SpecBranch would significantly stall parallel efficiency. Combining single-sequence parallelism and H-RAD alone cannot counteract rejections or reduce rollbacks. Early termination of the draft model would make the target model a bottleneck, leaving overall latency unchanged. In summary, SpecBranch is a unified framework with two synergistic modules. H-RAD first identifies unconfident tokens that are prone to rollback. Then the branch-parallel strategy counteracts rejections while accelerating inference in parallel.

| $\epsilon$ | Implicit(Confidence) | | Implicit(Entropy) | | Hybrid(H-RAD) | | Explicit(Feature) | |
|---|---|---|---|---|---|---|---|---|
| | M | Speedup | M | Speedup | M | Speedup | M | Speedup |
| 0.1 | 2.39 | 1.67× | 2.37 | 1.65× | 3.02 | 1.98× | 2.21 | 1.51× |
| 0.2 | **2.63** | **1.76×** | **2.51** | **1.74×** | **3.24** | **2.04×** | | |
| 0.4 | 2.31 | 1.59× | 2.27 | 1.56× | 3.19 | 2.02× | | |
| 0.6 | 1.98 | 1.47× | 1.96 | 1.45× | 2.97 | 1.95× | | |

Table 19:  The hyperparameter sensitivity on the branch-parallel structure.

## G   FURTHER ANALYSIS AND DISCUSSION

### G.1   MEMORY CONSTRAINED SCENARIOS

We have discussed the memory consumption of SpecBranch in Section 6.2. Here, we provide a more detailed discussion about memory-constrained scenarios.

**Clarification of the Application Scenarios**   First, we clarify the application scenarios of SpecBranch: most existing methods operate on a "draft-then-verify" sequential execution, which limits the ability to fully utilize the computational resources available. In contrast, the main application scenarios for SpecBranch focus on environments with sufficient computational resources to enable parallel frameworks, where serialized execution is not able to adequately leverage available resources.

**SpecBranch in Memory-Abundant Scenarios.**     Specifically, SpecBranch is well-suited for multi-GPU parallel scenarios in cloud environments with ample GPU resources, as well as for cloud-edge collaborative settings. In such scenarios, the draft and target models are deployed on the edge and cloud devices, respectively. Additionally, it can be applied to heterogeneous processor environments, including CPU/GPU configurations or heterogeneous GPUs. We foresee significant potential for SpecBranch in these scenarios, where both the draft and target models are independently deployed across processors. This avoids slowdown caused by memory contention. Our main experiments are conducted under these settings. Furthermore, integrating tensor parallelism (TP) with SpecBranch in these environments (Zhong et al., 2024) can further enhance acceleration in the future.

| Methods | MT Bench | QA | Summarization | Math | RAG | Translation | Avg. |
|---|---|---|---|---|---|---|---|
| Sps | 2.03× | 2.12× | 2.02× | 2.06× | 1,94× | 2.21× | 2.06× |
| SpecBranch | 3.02× | 2.87× | 2.95× | 3.21× | 2.76× | 3.28× | 3.02× |
| SpecBranch(PP) | 2.80× | 2.56× | 2.57× | 2.93× | 2.41× | 3.02× | 2.73× |
| **Performance retain** | **92.57%** | **89.03%** | **89.78%** | **91.38%** | **87.21%** | **91.93%** | **89.76%** |

Table 20: Comparisons of Deepseek 1.3B&33B on the Spec-Bench tasks with the proposed Spec-Bench in memory-constrained scenarios.

**SpecBranch in Memory-Constrained Scenarios.**     We also consider memory-constrained scenarios, where resource contention between the draft and target model may arise. To mitigate this, we consider a common real-world scenario using the A100 40GB GPU. In this case, a large target model (33B) is deployed across two GPUs, while a smaller draft model (1.3B) is deployed on one of these GPUs. In such a scenario, we employ a modified pipeline parallelism (PP) version of SpecBranch to alleviate resource contention based on PEARL (Liu et al., 2024b). Specifically, the target model's computation is sequential across multiple GPUs: while the target model runs on GPU 0, the draft model can operate in parallel on GPU 1 to generate the first $\lceil \frac{\gamma}{2} \rceil$ tokens; when the target model progresses to GPU 1, the draft tokens generated on GPU 1 are transferred to the idle GPU 0, which continues generating the remaining $\lceil \frac{\gamma}{2} \rceil$ tokens, thereby avoiding memory contention. Although this PP approach introduces additional communication overhead, it effectively mitigates memory contention in this specific scenario. We conduct experiments with Deepseek 1.3B&33B on Spec-Bench and find that SpecBranch maintains approximately **90%** of the reliable performance through PP, which outperforms the vanilla SD and auto-regressive decoding in memory-constrained settings. The results are shown in Table 20.

Meanwhile, SpecBranch conducts extensive experiments on poorly aligned lightweight model combinations (68M&7B, 68M&13B). The experimental results demonstrate that, under H-RAD's rollback mitigation, SpecBranch exhibits better adaptability and energy efficiency for small draft models, making it more suitable for deployment in resource-constrained environments.

| Methods | MT Bench | QA | Summarization | Math | RAG | Translation | Avg. |
|---|---|---|---|---|---|---|---|
| PEARL(Sps) | 1.74× | 1.64× | 1.70× | 1.55× | 1.56× | 1.65× | 1.64× |
| **SpecBranch *w/o branch*** | **1.87×** | **1.73×** | **1.75×** | **1.71×** | **1.73×** | **2.03×** | **1.81×** |

Table 21: Comparisons of Vicuna 68M&13B on the Spec-Bench tasks with the proposed Spec-Bench in single GPU scenarios.

**SpecBranch in Single GPU Scenarios.**     In the extreme resource-constrained scenarios, where only a single GPU is available for inference deployment, we can still apply the pipeline parallelism (PP) strategy by offloading the draft model to the CPU (DRAM), enabling heterogeneous parallelism between the CPU and GPU. This approach is left as future work for SpecBranch. If deployment is limited to a single GPU, SpecBranch degenerates to a non-parallel framework, as discussed in Section 6.2 under SpecBranch *w/o branch*. In this case, the H-RAD component operates independently of the parallel framework and can be seamlessly integrated with the existing draft-then-verify methods. In contrast, PEARL's pre-verify and post-verify stages degenerate to vanilla SD under extreme resource constraints. We conduct experiments with Vicuna 68M and 13B on Spec-Bench using a single A100 GPU and find that SpecBranch *w/o branch* outperforms PEARL (vanilla SD) in these single-GPU scenarios. The results are shown in Table 21.

## G.2 TREE STRUCTURE AND TEMPORAL MISMATCH

In this section, we provide more detailed discussions about the tree structure and temporal mismatch to gain an in-depth understanding of SpecBranch (Section 5).

**Tree-based Structure** We primarily focus on the vanilla tree structures. For example, SpecInfer (Miao et al., 2024) constructs a token tree using $k$ independent sequences, a topology that is constrained by the expected number of tokens it can accept, regardless of the tree size, as shown in Fig. 18(a). However, this structure is dense, as top-$k$ sampling is applied to each token, which generates additional sequences. In the case where the draft length is $\gamma$ and the tree size is $k$, the number of tokens for each round is given by $\frac{k^\gamma - 1}{k-1}$. We observe that the number of tokens in a tree structure grows exponentially with $\gamma$, causing KV-Cache storage to increase exponentially as well. To address this challenge, EAGLE2 (Li et al., 2024b) and SEQUOIA (Chen et al., 2024) employ dynamic draft tree adjustments to prune unnecessary branches, resulting in a sparse tree structure. Our SpecBranch adopts a similar sparse branch structure. The difference is that, unlike traditional tree structures, where each token generates a branch, our approach utilizes H-RAD to predict high-impact token positions and spawns sparse branch points that minimize speculative divergence. As illustrated in Fig. 18(b), we branch at $b \leq \gamma$ positions, resulting in the number of tokens per round in the branch structure being $k \cdot \gamma - (k-1) \cdot (b+1)$, which is significantly smaller than in vanilla tree structures. In general, vanilla tree structures maintain the KV-Cache for the entire search tree, with a space complexity of $O(k^\gamma)$, whereas SpecBranch reduces the complexity to $O(k \cdot \gamma)$ by sparsely targeting specific branches.

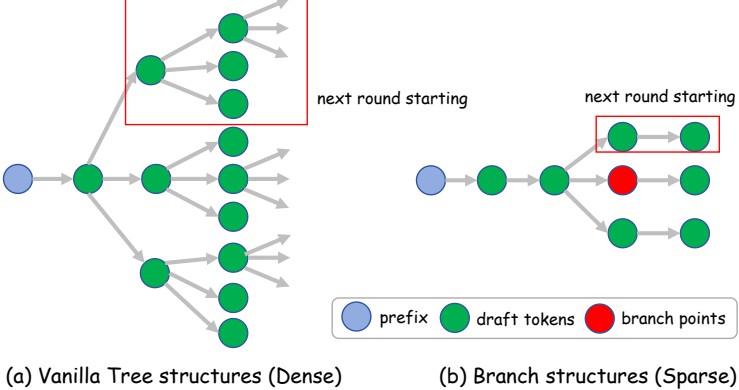

**(a) Vanilla Tree structures (Dense)**          **(b) Branch structures (Sparse)**

Figure 18: Comparison of the vanilla tree structures (dense) and branch structures (sparse).

Moreover, trees require complex attention mask verification to process the entire structure in parallel, whereas our approach only needs to verify the branch point. From a practical perspective, branch structures offer advantages in terms of deployment complexity and resource efficiency. Additionally, for sparse tree structures, SpecBranch can be easily integrated, which we leave for future work. However, a key limitation is that tree structures are not well-suited for parallel architectures as illustrated next.

**Temporal Mismatch** As shown in Fig. 19, we observe that the parallel branch stage introduces a temporal mismatch between drafting and verification. In the draft stage (a), the previous tokens have already been verified by the target model (except for the first round, where the target model lacks feature information). As a result, H-RAD can utilize the historical target model feature pairs to predict the token generation length for the next round. In this stage, H-RAD uses features in a priori fashion for prediction. However, in the branch stage (b), the draft model proactively generates speculative branches concurrently with target model verification. Such parallelization of drafting and verification leads to a unique challenge that tokens from the previous round have not been fully verified by the target model before new tokens are generated. This temporal mismatch prevents H-RAD from obtaining reliable features from the target model before new branches generate tokens.

For vanilla tree structures, temporal mismatch limits the verification process to only determining the first token of the next round. In other words, the next round starting (Fig. 18) in tree structures is not just one token, but rather $(\gamma - 1) \cdot k$ tokens from a partial tree. This means that during the parallel

phase, the number of tokens in a dense tree structure will reach $(\gamma - 1) \cdot k \cdot \frac{k^\gamma - 1}{k - 1}$, which is $(\gamma - 1) \cdot k$ times higher than the KV-Cache in the previous intermediate steps, further exacerbating the memory overhead. On the other hand, branch structures only retain one branch after each verification step due to sparsity. This ensures that each new round starts afresh and mitigates the memory overhead.

Temporal mismatch is a unique "byproduct" of parallelism, which is unavoidable. Its mistreatment would delay the validation of draft tokens by the target model, potentially forming bottlenecks without specialized treatment such as PEARL. In our work, SpecBranch explicitly addresses such temporal mismatch by introducing adjustments to mitigate such asynchronous challenge. This is done via a posterior approach for token selection in case the target model's features are unavailable. In the next Section, we also explore priori methods, which resolve this issue and enable feature reuse similar to EAGLE.

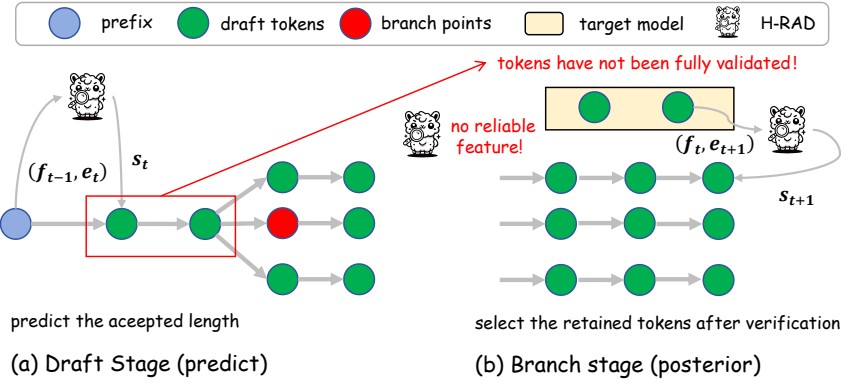

Figure 19: Comparison of the Draft and Branch Stages. The parallel branch stage introduces a temporal mismatch between drafting and verification.

**Posterior Approach**   To address this temporal mismatch, we introduce a posterior approach in Section 5.2. Once the previous tokens are fully verified by the target model, since $\gamma$ represents the speed ratio $c$ between the draft and target models, by the time this happens, the branch has also completed token generation. Then, for the remaining branch $V$, we use the features $(f_t, e_{t+1})$ from the current round as input to H-RAD and select the retained tokens $\mathcal{H}_t$ after the verification step. This posterior approach effectively resolves the temporal mismatch between verification and drafting, ensuring that H-RAD always uses the most up-to-date and relevant context. It also leverages parallelism, making the time loss from the posterior approach negligible. However, on the other hand, unlike in the draft stage, we cannot implement early stopping. While the parallelization mostly mitigates time impact due to bottlenecks at the target model, this introduces additional token waste. Moreover, this feature utilization method does not fully align with the current mainstream methods such as EAGLE (Li et al., 2024a).

**A Priori Approach**   We further explore a priori methods to unify the verification processes between the draft and branch stages. We leverage the *temporal locality* of transformer hidden states, where features from earlier time steps, though slightly outdated, still retain sufficient predictive power. Thus, H-RAD can proactively reuse features from the target model, specifically $(f_{t-1}, e_t)$, to predict $\mathcal{H}_t$, whereas the posterior approach uses $(f_t, e_{t+1})$ as input. In other words, if we cannot reuse features from the previous round's token, we rely on token features from the two rounds back, which exhibits a temporal decay, where the predictive capability of the target model features diminishes across multiple rounds.

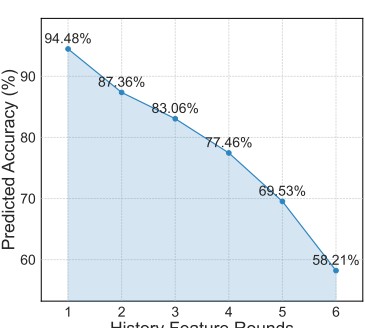

Figure 20: Predictive capability of the features progressively decay of LLaMA 68M&7B on HumanEval.

To quantify such decay, we conduct an evaluation of LLaMA 68M&7B on HumanEval as shown in Fig. 20. It measures the prediction accuracy of the MLP used in H-RAD. We observe that as the distance and number of rounds increase – transitioning from $(f_t, e_{t+1})$ to $(f_{t-n}, e_{t+1-n})$ – the contextual informa-

tion and predictive capability of the features decay gradually. However, note that the features from the previous two rounds $(f_{t-1}, e_t)$ still exhibit some predictive capacity. In scenarios where speed is not as critical, these stale features may serve as a viable surrogate for the more recent features $(f_t, e_{t+1})$. This a priori approach better unifies the operations of the draft and branch stages. While the acceleration effect is not as significant as the posterior approach, the early stopping mechanism reduces the number of tokens generated. Additionally, this method can be combined with EAGLE (Li et al., 2024a) to reuse features with the draft model in future.

### G.3 MORE DISCUSSIONS ABOUT OUR CONTRIBUTION

The key novelty/difference from the prior methods is that Prior Work such as PEARL is **Rollback-Oblivious** and **Static**, but our approach is **Rollback-Aware** and **Dynamic**. The previous works do not take any active measures as the rollback-oblivious methods would continue processing those "doomed tokens" and make the target model a bottleneck with useless computations, especially under poorly aligned models (shown in Fig. 5). In sharp contrast, our approach makes it preemptive, since we are hedging against possible failures. This is accomplished by:

✧ **Anticipates Failure:** At points of high uncertainty, we proactively spawn alternative futures via "branch resampling", i.e., we do not just hope a single path to succeed; we plan for its potential failure.

✧ **Instantly Prunes Bad Paths:** Once the branch point is verified, we commit to the correct path and immediately terminate the invalid ones, ceasing all computation and freeing their KV-Cache. This is a fundamental difference from the static pipeline of prior work.

✧ **Maintains Sparsity and Efficiency:** Unlike tree-based methods (e.g., SpecInfer) that create a dense, memory-intensive tree structure, our branches are more sparse, created only at critical junctures. This reduces KV-Cache complexity substantially, making our parallel approach practical and scalable (Appendix G.2, Fig. 18).

Furthermore, the second novelty is the unification of implicit and explicit drafting strategies since they have their own pros and cons. The hybrid approach we proposed makes the drafting problem simpler and more tractable: 1) reducing the difficult $N$-class length prediction into a 3-class problem where features are separable; 2) using the explicit methods for what it is good at by identifying the high-confidence hard signals of the two extreme cases, but for ambiguous "uncertain" cases where the MLP falters, we fall back to the implicit confidence score as a "soft signal". In fact, it turns out that such a simple and elegant solution is effective and would be able to cut the rollback rates by up to 50% based on our experimental validations.

The key differences between the existing architectures are highlighted by Table 1 (Sec. 2), which states that SpecBranch and PEARL are the only two parallel drafting techniques, but SpecBranch takes a step forward to make the process rollback-aware.

### G.4 MORE DISCUSSIONS ABOUT TRAINING-FREE

SpecBranch is the first parallel framework with hybrid drafting structures that **DOES NOT require additional training of draft models**. Similar to the previous works, we define training as an extra procedure with a non-trivial cost. For example, REST (He et al., 2023) replaces the draft model with a trained retrieval model, advancing the development of Self-SD. AdaDecode (Wei et al., 2025) introduces lightweight language model (LM) heads at intermediate layers, achieving a more efficient layer skip compared to truly "training-free" methods like SWIFT (Xia et al., 2024a). In contrast, the training cost of a three-layer MLP for our predictor module is indeed lightweight.

### G.5 MORE DISCUSSIONS ABOUT COMPUTE AND COMPLEXITY

**Real-time Compute** While branch resampling accelerates wall time, it also introduces additional computational overhead. However, current LLM inference acceleration is more constrained by memory bounds. In contrast to Autoregressive Decoding, Speculative Decoding mitigates this limitation by increasing computational resources. Therefore, increasing computation to achieve higher speedups shows significant potential. SpecBranch fully leverages hardware resources to provide higher speedups.

On the other hand, SpecBranch uses H-RAD to strategically spawn branches, which reduces token rollback by 50%, and the additional energy consumption remains lower compared to PEARL. Note that we have considered the energy cost in this work (see Section 6.2 and Appendix F.4) which has not been unveiled by contemporary works and we believe this is essential for real-world production environments.

**Complexity** In fact, the additive complexity turns out to be worth it. This is because the performance gains outweigh the cost: our system as a whole deliver $1.8\times - 4.5\times$ speedups with 50% less rollback, while the cost can be easily justified: 1) H-RAD is a lightweight 3-layer MLP (trained offline in 5 minutes on a single A100) with minimal overhead-predict latency (0.38% of total latency), requiring no additional draft-model training; 2) Branch resampling uses adaptive logic to auto-tune branch counts based on draft tokens confidence, that alleviates manual hyperparameter tuning. For services like ChatGPT, Google's Bard/Gemini, or a large enterprise API, inference is a massive operational expense. If the performance margin compared to existing works could translate directly into millions of dollars in savings on GPU infrastructure and energy costs, rather than being a barrier to widespread adoption. The implementation of our framework is not complex either as we will make it publicly available later. The core software code only contains about 500 lines of Python, in which each component is modularized and easy to adapt to other techniques.

### G.6 GROUP SPECULATIVE DECODING AND HETEROGENEOUS DEVICES

**Group SD within a Single GPU Cluster** In SpecBranch, we rigorously validate the acceleration effectiveness for a single draft-target model pair. However, as Model-as-a-Service (MaaS) becomes more prevalent, serving multiple model instances within a cluster is increasingly important for maximizing resource utilization. Meanwhile, we plan to extend our system to support simultaneous SD for multiple draft-target pairs with more than one draft model per GPU and high throughput. For example, in an $8\times$A100 80G configuration, we can deploy 7 different target models (33B) on 7 GPUs and up to 15 draft models (1.3B) on the remaining GPUs. These draft models can be paired with the same or different target models through advanced communication and network design.

**Speculative Decoding on Heterogeneous Devices** During validation, we find that large models often require multiple high-end GPUs to fit into memory (e.g., a 70B-parameter model experiment requires at least $4\times$A100 40G GPUs). While one solution discussed in the previous section is to pack multiple draft models in a single device, the prerequisite for large-memory devices remains unchanged. However, we discover that under the setting of SD, homogeneous deployment is unnecessary, i.e., the draft models do not need to be co-located on the same device as the target models. Thus, we present a heterogeneous structure that runs smaller draft models on consumer-grade GPUs (e.g., RTX 3090) and connects them over the network to target models on data-center GPUs (e.g., A100). The challenges on the system level, including synchronization, communication latency, and workload balancing, remain as future work for SpecBranch.

### G.7 DETAILED ALGORITHM PERIOD AND DISCUSSION OF SPECBRANCH

While SpecBranch shares the fundamental parallelism principle with PEARL, it transcends a simple combination of "PEARL + MLP." Instead, it is a synergistic co-design of the lightweight H-RAD module within a rollback-aware parallel framework. Unlike PEARL, which neglects the cost of token rollbacks, SpecBranch explicitly addresses this limitation through two core architectural innovations:

⬦ **Active Topology Control:** The H-RAD module does not merely passively judge tokens; it actively governs *when* to branch, *where* to locate the branch point, and *how many* branches to spawn. This transforms the draft–verify pipeline from a static parallel scheme into a dynamic, uncertainty-adaptive process.

⬦ **Rollback Awareness:** By explicitly modeling rollback risks, SpecBranch restricts branching to high-uncertainty points and verifies specific branch points rather than the entire tree. This allows the system to immediately terminate those "doomed branches" upon verification failure, reducing rollback tokens by up to 50% compared to PEARL.

The detailed workflow of the SpecBranch algorithm is described below:

**(1) Drafting Phase.** Given a context prefix $X_{1:t-1}$, the draft model generates a sequence of candidate tokens. Simultaneously, H-RAD analyzes the multi-layer target features (cached from the previous decoding step) to assess the entropy of the current generation. H-RAD classifies the current position into three distinct regions: (i) *all-accept* (high confidence), (ii) *all-reject* (certain failure), or (iii) a confidence-based *soft region* (uncertainty).

**(2) Branch Point Selection.** If the generation falls into the *soft region*, H-RAD marks the current token $x_b$ as a *branch point*. At this juncture, the drafting process splits:

- The stable prefix $X_{1:b-1}$ is immediately sent to the target model for verification.
- Meanwhile, the draft model samples the top-$k$ alternatives for $x_b$ and continues drafting $k$ parallel branches. Crucially, all branches share the same KV-cache for the prefix $X_{1:b-1}$ to minimize memory overhead.

**(3) Prefix Verification.** Executed in parallel with branch drafting, the target model verifies the prefix segment $X_{1:b-1}$. If any token within this segment is rejected, the system identifies a critical failure. All pending parallel branches are immediately discarded, and the process falls back to standard speculative decoding resampling (similar to the rollback mechanism in PEARL) to ensure correctness.

**(4) Branch Verification and Selection.** If the prefix $X_{1:b-1}$ is fully accepted, the target model computes the probability distribution $p(x_b)$ at the branch point. We then apply a *Branch Speculative Sampling* strategy (detailed in Appendix D) to the set of $k$ branch candidates $\{x_b^i\}$, utilizing both the target distribution $p$ and the draft distribution $q$ to ensure the final output matches the target distribution. As soon as one candidate is accepted, SpecBranch retains the corresponding path and discards all other branches (along with their KV-caches). Consequently, the target model only ever continues with a **single** valid sequence, eliminating the need for complex tree attention masks.

**(5) Posterior Drafting.** Upon selecting the retained branch, the system updates the KV-cache and target features. H-RAD is then invoked again to determine the next draft length and identify potential future branch points, restarting the iterative process.

## G.8 EXTENDED ANALYSIS ON DRAFT–TARGET MODEL SIMILARITY

To comprehensively assess the generalization capability of SpecBranch, we conducted extensive experiments across a wide spectrum of draft–target similarity levels. Our evaluation benchmark spans from weakly aligned pairs (e.g., LLaMA-68M/7B, Vicuna-68M/13B) to strongly aligned configurations (e.g., DeepSeek-1.3B/33B, LLaMA-3.1-8B/70B). We observe that SpecBranch exhibits a dynamic adaptation mechanism where distinct components dominate performance depending on the alignment regime:

- ◇ **Low-Similarity Regimes:** In scenarios with weak alignment, the performance gain is primarily driven by the rollback-aware H-RAD mechanism. By effectively identifying and terminating invalid branches early, H-RAD significantly mitigates the latency penalty associated with frequent corrections. Empirically, this results in a reduction of rollback rates by approximately 50% compared to PEARL, ensuring stable parallelism even when the draft model provides lower-quality predictions.
- ◇ **High-Similarity Regimes:** Conversely, for well-aligned model pairs, the contribution of Branch Resampling becomes dominant. The naturally high acceptance rate allows the parallel branches to be validated successfully more often, thereby maximizing token throughput and fully leveraging the system's parallel capacity.

These findings confirm that SpecBranch is not limited to specific model combinations but automatically adapts to the underlying model alignment, maintaining a consistent performance advantage over baselines across diverse inference conditions.

## G.9 HIGH-BATCH PERFORMANCE ANALYSIS

**High-Batch Scalability** To bridge the gap towards industrial deployment, we are currently integrating SpecBranch into the nano-PEARL framework (Liu et al., 2024b), a lightweight, high-concurrency

engine based on nano-vllm, while conducting pilot studies with industrial partners. Addressing the critical requirement for high-throughput scenarios, we further compare SpecBranch against EAGLE3 (Li et al., 2025) on the Qwen3-32B model across varying batch sizes ($bs$). As shown in Table 22, EAGLE3 exhibits severe performance degradation as concurrency increases, dropping to near-baseline speeds ($\sim 1.05\times$) at $bs = 32$. This bottleneck stems from its tree-based design, where the computational load scales linearly with tree size $\times$ batch size, rapidly saturating hardware arithmetic intensity. In contrast, SpecBranch mitigates this by selectively branching only on low-confidence tokens and delaying compute saturation via a reduced $k_{max}$. Furthermore, by employing a lightweight verification step that bypasses the complex tree attention masks inherent to prior methods, SpecBranch demonstrates superior robustness in high-load environments.

While mainstream speculative decoding (SD) research primarily benchmarks at batch size $bs = 1$, industrial deployments require high throughput. In high-batch regimes, the system bottleneck shifts from memory bandwidth (memory-bound) to arithmetic intensity (compute-bound). This transition poses a fundamental challenge for the existing SD methods: as $bs$ increases, the additional computation required for drafting and verification often saturates hardware resources, nullifying the speed gain of SD unfortunately. For instance, state-of-the-art tree-based methods like EAGLE3 is found to degrade to baseline speeds at $bs > 8$. To demonstrate SpecBranch's industrial viability, we conduct a comprehensive high-batch evaluation.

**Implementation Details**  We integrate SpecBranch into **nano-PEARL** (Liu et al., 2024b), a lightweight, high-concurrency inference engine based on nano-vllm. This framework supports advanced optimizations including PagedAttention, FlashAttention, and CUDA graphs.

- **Hardware:** Experiments are conducted on $3\times$ NVIDIA A100-80G (NVLink) GPUs.

- **Configuration:** We employ Tensor Parallelism (TP). For the draft model (Qwen3-0.6B), we set TP=1; for the target model (Qwen3-32B), we set TP=2.

- **Parameters:** To balance computational load in high-concurrency settings, SpecBranch is configured with a reduced maximum branch count of $k_{max} = 3$. The EAGLE3 baseline utilizes TP=2 following its official implementation. All baselines run with CUDA graphs enabled.

| Dataset | Methods | Batch Size (Qwen3-32B) | | | | |
| --- | --- | --- | --- | --- | --- | --- |
| | | batch = 1 | batch = 4 | batch = 8 | batch = 16 | batch = 32 |
| **HumanEval** | EAGLE3 | $1.98\times$ | $1.67\times$ | $1.34\times$ | $1.13\times$ | $1.05\times$ |
| | **SpecBranch** | **$2.34\times$** | **$2.11\times$** | **$1.97\times$** | **$1.74\times$** | **$1.55\times$** |
| **GSM8K** | EAGLE3 | $2.03\times$ | $1.74\times$ | $1.43\times$ | $1.15\times$ | $1.07\times$ |
| | **SpecBranch** | **$2.57\times$** | **$2.34\times$** | **$2.06\times$** | **$1.88\times$** | **$1.63\times$** |

Table 22: High-Batch performance comparison on Qwen3-32B. SpecBranch maintains robust acceleration under high concurrency ($bs = 32$) compared to EAGLE3.

**Results Analysis**  As shown in Table 22, we observe a distinct divergence in performance scaling. EAGLE3's acceleration collapses as batch size increases, dropping to near-negligible levels ($1.05\times$) at $bs = 32$. In contrast, SpecBranch maintains robust acceleration, achieving $\geq 1.55\times$ speedup even at $bs = 32$. This performance gap is driven by two critical design advantages:

✧ **Selective vs. Dense Drafting:** EAGLE3 employs a static tree structure where the computational load scales multiplicatively with Tree Size $\times$ Batch Size, causing rapid hardware saturation. SpecBranch, conversely, utilizes an entropy-based selective strategy. By spawning parallel branches *only* at high-uncertainty tokens and capping $k_{max}$, SpecBranch effectively manages the total FLOPs, delaying the onset of the compute-bound regime.

✧ **Lightweight Verification:** High-batch verification introduces significant latency for the target model. Tree-based methods suffer from the overhead of processing complex tree attention masks across the entire batch. SpecBranch bypasses this bottleneck by validating only specific, sparse branch points. This streamlined verification process minimizes the pre-fill latency penalty, making it inherently more scalable for high-throughput environments.

These results confirm that SpecBranch offers a competitive, robust solution for industry-scale deployment, effectively bridging the gap between theoretical SD gains and practical high-concurrency demands.

### G.10 EXTENDED COMPARISON WITH EAGLE-3

While comparisons with EAGLE and EAGLE-2 are detailed in Appendix F.1, we extend our evaluation to include the recently proposed EAGLE-3 (Li et al., 2025). Due to EAGLE-3's lack of official support for the LLaMA-2 architecture, we conduct these comparative experiments using the Qwen3-32B model. Both implementations utilize the Transformers-based architecture for a fair comparison.

| Model | Methods | HumanEval | | GSM8K | | CNN/DM | | Speed | Avg. |
|---|---|---|---|---|---|---|---|---|---|
| | | M | Speedup | M | Speedup | M | Speedup | (tok/s) | Speedup |
| Qwen3 | EAGLE-3 | 2.88 | 2.16× | 3.17 | 2.36× | 2.63 | 2.07× | 28.61 | 2.20× |
| | **SpecBranch** | **5.62** | **2.57×** | **7.64** | **2.69×** | **4.96** | **2.35×** | **33.02** | **2.54×** |

Table 23: Comparison with EAGLE-3 on Qwen3-32B. SpecBranch consistently outperforms EAGLE-3 across all datasets despite Qwen3's large vocabulary size.

As shown in Table 23, SpecBranch consistently outperforms EAGLE-3 across all datasets. Notably, EAGLE-3 exhibits limited acceleration on the Qwen3 model. We attribute this performance bottleneck to Qwen3's extensive vocabulary size, which imposes significant computational overhead during the tree verification phase inherent to EAGLE's design. In contrast, SpecBranch's sparse branching strategy remains efficient in large-vocabulary regimes. We acknowledge that EAGLE-3's use of a dedicated draft model is a significant but orthogonal design choice. SpecBranch currently utilizes a standard draft model; however, this opens a promising avenue for future work to integrate on-policy online distillation, developing specialized draft models to further enhance the parallel H-RAD framework.

## H USE OF LLMS STATEMENT

We do not use any Large Language Models (LLMs) in paper writing.

