# OpenReview forum: "SpecBranch: Speculative Decoding via Hybrid Drafting and Rollback-Aware Branch Parallelism"
_ICLR.cc/2026/Conference — ICLR 2026 Poster_

### Official Review · Reviewer_oJJE · 2025-10-22

**Soundness:** 3
**Presentation:** 4
**Contribution:** 3
**Rating:** 8
**Confidence:** 4

**Summary:**

The method introduces parallel speculative branches, adaptively chooses draft lengths, and reuses features of the target model to hedge against incorrect drafts and reduce rollback tokens.

**Strengths:**

The idea of modelling speculative decoding as analogous to branch prediction in CPUs is clever, and the introduction of parallel speculative branches is conceptually interesting.

**Weaknesses:**

It is not fully clear how generally applicable the method is across diverse architectures, tasks, or deployment settings.

**Questions:**

I suggest adding ablation experiments exploring branch length, number of speculative branches, draft vs. target model similarity, and rollback frequency under different task/benchmark settings.

---

> ### Author Response · Authors · 2025-11-19
> **Response to Reviewer  oJJE (Part 1 / 3)**
>
> We sincerely appreciate your insightful and professional comments. We are delighted by your recognition of our core idea, modelling speculative decoding as analogous to branch prediction in CPUs. We are also truly grateful for your remarks on the paper’s clarity, noting that our key ideas are presented with sufficient detail and that SpecBranch offers meaningful *novelty*  in parallel speculative branches compared to prior speculative decoding methods. Your recognition of these aspects means a lot to us, serving as **a beacon of encouragement** that motivates us to continue refining and advancing this line of research.
>
> In addition to your acknowledgement of our paper’s main contributions, we have carefully considered your constructive feedback and questions with more ablation experiments. Specifically, we have included comprehensive results in the Appendix covering cross-task generalization (Appendix E.4), comparisons with tree-based methods like EAGLE (Appendix F.1), $k\_{max}$ analysis (Appendix F.2), latency stability (Appendix F.3), and rollback ratios (Appendix F.6), alongside memory-constrained applications (Appendix G.1) and further ablations (Appendix F.8, F.10).
>
> These findings strongly validate SpecBranch's robustness across diverse architectures, tasks, and deployment settings. To fully address your concerns, we provide **a more detailed discussion and experiments below**.
>
> ***
>
> ***Q1： I suggest adding ablation experiments exploring branch length, number of speculative branches, draft vs. target model similarity, and rollback frequency under different task/benchmark settings.***
>
> A1: We sincerely appreciate your valuable comments. Here, we provide further supplementary results and detailed discussions regarding the specific experiments you mentioned.
>
> 1. **Branch length**
>
> Thank you for the question. We define the branch length as the default draft length \\(\gamma\\) in each decoding round. Unlike PEARL\[1], which fixes \\(\gamma = c\\) (the theoretical speedup ratio) for all model pairs, SpecBranch introduces a rollback-aware parallel SD architecture. This makes the optimal \\(\gamma\\) depend on the acceptance–rollback trade-off predicted by our theoretical model. We evaluate different \\(\gamma\\) values on HumanEval for two representative cases:
>
> \- LLaMA 68M & 7B: low draft–target alignment, \\(c = 10\\)
>
> \- LLaMA 3.1 8B & 70B: high alignment, \\(c = 5\\)
>
> > R3-Table1: Experiments of overall speedup and mean accept length for SpecBranch under different branch lengths
>
> | **γ** | **LLaMA 68m -7B（c=10）** |           | **γ** | **LLaMA 3.1 8B-70B(c=5)** |           |
> | ----------- | ----------------------- | --------- | ----------- | ------------------------- | --------- |
> |             | M                       | Speedup   |             | M                         | Speedup   |
> | 5           | 2.39                    | 1.69x     | 3           | 14.37                     | 3.45x     |
> | 6           | 2.63                    | 1.87x     | 4           | 17.51                     |  3.74x    |
> | 7           | 2.97                    |  1.96x    | **c=5**     | **21.74**                 | **4.02x** |
> | **8**       | **3.24**                | **2.04x** | 6           | 16.89                     | 3.65x     |
> |  9          | 3.01                    | 1.98x     | 7           | 13.98                     | 3.42x     |
> | c=10        | 2.87                    | 1.93x     | 8           | 12.45                     | 3.06x     |
>
> **For low-similarity draft–target pairs(LLaMA)**, we observe that the optimal branch length does not coincide with the speed ratio $c=10$, but instead occurs at $\gamma=8$. This observation is consistent with the trade-off analysis presented in Figure 2 and Theorem 1: when token acceptance rates are low, the latency penalty associated with rollbacks becomes the dominant factor, thereby diminishing the gains from parallelism. Consequently, the optimal branch length falls strictly below $c$—a critical nuance that PEARL overlooks.
>
> **For high-similarity pairs (LLaMA 3.1)**, the behavior of the optimal branch length aligns with PEARL's findings. When the semantic similarity between the draft and target models is high, the benefits of parallelism dominate the trade-off. Therefore, as $\gamma$ approaches $c$, thereby increasing the degree of parallelism, the overall speedup rises correspondingly.

---

> > ### Author Response · Authors · 2025-11-19
> > **Response to Reviewer oJJE (Part 2 / 3)**
> >
> > 2. ***Number of speculative branches***
> >
> > Thanks for the sharp catch. The number of speculative branches $k$ is a crucial hyperparameter. As shown in Section 6.2 and Appendix G.2, memory usage increases only mildly when moving from 1 to 6 branches. To further analyze its effect, we ran additional experiments on HumanEval for both Vicuna 68M→13B and LLaMA-3.1-8B→70B. Results are summarized in R3-Table 2.
> >
> > > R3-Table2: Experiments of overall speedup and rollback rate for SpecBranch under varying $ k\_{max}$
> >
> > | **$ k\_{max}$** | **1**  |         | **2**  |         | **4**  |         | **6**     |           | **12**     |                 | **18** |         |
> > | --------------------- | ------ | ------- | ------ | ------- | ------ | ------- | --------- | --------- | ---------- | --------------- | ------ | ------- |
> > | **Model**             | RB     | Speedup | RB     | Speedup | RB     | Speedup | RB        | Speedup   | RB         | Speedup         | RB     | Speedup |
> > | Vicuna 68M&13B        | 87.56% |  2.05x  | 62.13% | 2.16x   | 48.02% | 2.28x   | 45.14%    | 2.36x     | **39.60%** | **2.47x** | 38.92% | 2.48x   |
> > | LLaMA 3.1 8B&70B      | 19.63% | 3.76x   | 15.63% | 3.88x   | 11.63% | 3.97x   | **9.51%** | **4.02×** | 8.93%      | 4.05×           | 8.87%  | 4.06×   |
> >
> > Across both model families, we observe a consistent trend: Increasing $k\_{max}$ initially yields rapid speedup gains, which then gradually saturate. Rollback rate decreases monotonically as more branches provide more opportunities to correct uncertain tokens early.
> >
> > **For low-similarity draft–target pairs**, acceptance rates are low, so using more branches helps reduce rollback and improves parallel efficiency—though with slightly higher memory cost.
> >
> > **For high-similarity pairs**, acceptance is already strong, so only a small number of branches is sufficient to reach the optimal trade-off.
> >
> > *The number of speculative branches* provides a flexible knob for balancing speedup, rollback reduction, and memory usage, and we will include these additional results in the final manuscript.
> >
> > ***
> >
> > 3. ***Rollback frequency under different task/benchmark settings.***
> >
> > As discussed in Section 6 and Appendix F.6, rollback is an important metric for parallel efficiency. Here, we provide more evaluation results on HumanEval, GSM8K, and CNN/DM, as shown in the Table below.&#x20;
> >
> > > R3-Table3: Comparison of Rollback Rates for Vicuna 68M&13B
> >
> > | **Methods**   | **Sps** |       | **AdaEDL** |       | **PEARL** |       | **SpecBranch** |          |
> > | ------------- | ------- | ----- | ---------- | ----- | --------- | ----- | -------------- | -------- |
> > |               | M       | RB(%) | M          | RB(%) | M         | RB(%) | M              | RB(%)    |
> > | **HumanEval** | 2.87    | 76.6  | 2.77       | 62.1  |  3.11     | 90.3  | **3.69**       | **46.8** |
> > | **GSM8k**     | 2.54    |  78.2 | 2.46       | 61.9  | 2.83      | 91.5  | **3.29**       | **47.2** |
> > | **CNN/DM**    | 2.07    | 89.7  | 2.01       | 70.5  | 2.89      | 94.9  | **3.21**       | **48.9** |
> >
> > > R3-Table4: Comparison of Rollback Rates for LLaMA-3.1 8b&70b
> >
> > | **Methods**   | **Sps** |       | **AdaEDL** |       | **PEARL** |       | **SpecBranch** |          |
> > | ------------- | ------- | ----- | ---------- | ----- | --------- | ----- | -------------- | -------- |
> > |               | M       | RB(%) | M          | RB(%) | M         | RB(%) | M              | RB(%)    |
> > | **HumanEval** | 5.25    | 14.9  | 4.96       | 12.2  | 17.28     | 44.2  | **21.74**      | **21.5** |
> > | **GSM8k**     | 5.15    |  16.8 | 4.97       | 13.8  | 14.33     | 29.0  | **18.08**      | **14.1** |
> > | **CNN/DM**    | 5.09    | 18.1  | 4.85       | 14.2  | 7.51      | 49.6  | **9.41**       | **25.8** |
> >
> > The results demonstrate that SpecBranch achieves significantly lower rollback ratios across datasets and subtasks than other methods. This is consistent with our prior justification that the H-RAD module plays an essential role in mitigating rollback. The additional evaluations in Fig. 10 further validate H-RAD's generalization across different tasks. It is worth mentioning that for poorly aligned model combinations (e.g., Vicuna, LLaMA), SpecBranch reduces the rollback ratio by nearly 50% compared to PEARL. Even for better-aligned model pairs (e.g., Deepseek, LLaMA-3.1), it achieves about a 10% reduction. These results indicate the potential of SpecBranch in resource-constrained environments where the draft model sizes are typically restrained.

---

> > > ### Author Response · Authors · 2025-11-19
> > > **Response to Reviewer oJJE (Part 3 / 3)**
> > >
> > > 5. ***Draft vs. Target model similarity***
> > >
> > > We thank the reviewer for this valuable suggestion. We have already conducted experiments across a wide range of draft–target similarity levels, from weakly aligned pairs (e.g., LLaMA-68M&7B, Vicuna-68M&13B) to strongly aligned ones (e.g., DeepSeek-1.3B&33B, LLaMA-3.1-8B&70B). These results are summarized in Figure 6 and Appendix E. Across all these settings, we observe the following consistent trends:
> > >
> > > **Low-similarity pairs:** SpecBranch benefits primarily from the rollback-aware H-RAD mechanism, which terminates invalid branches early and significantly reduces rollback rates (up to \~50% vs. PEARL), leading to more stable parallelism.
> > >
> > > **High-similarity pairs:** SpecBranch Resampling contributes more strongly, as the high acceptance rate allows parallel branches to sustain higher throughput.
> > >
> > > These findings confirm that SpecBranch automatically adapts to different alignment regimes and maintains an advantage over PEARL and other baselines under both weak and strong similarity. We will clarify this generalization behavior further in the revised manuscript.
> > >
> > > ***
> > >
> > > We hope our responses have effectively addressed your concerns and we are more than happy to include all these discussions in the camera-ready version of this work.

---

> ### Author Response · Authors · 2025-11-25
>
> Dear Reviewer,
>
> I hope this message finds you well. As the discussion period is nearing its end, I wanted to ensure we have addressed all your concerns satisfactorily. If there are any additional points or feedback you'd like us to consider, please let us know. Your insights are invaluable to us, and we’re eager to address any remaining issues to improve our work.
>
> Thank you for your time and effort in reviewing our paper!

---

### Official Review · Reviewer_MCa5 · 2025-10-31

**Soundness:** 3
**Presentation:** 3
**Contribution:** 3
**Rating:** 4
**Confidence:** 5

**Summary:**

Based on PEARL's parallel framework, SpecBranch introduces the Hybrid Rollback-Aware Draft Structure (H-RAD) and ingeniously integrates the branch resampling method, enabling branch-parallel execution of speculative decoding (SD). This design effectively addresses the serialized waiting bottleneck and high rollback issue inherent in traditional SD methods. The authors rigorously validate and elaborate on these two core innovations through comprehensive theoretical analyses—including Theorem 1 on latency under rollback and the truncated geometric distribution of accepted tokens—and extensive experimental evaluations.

**Strengths:**

1.	Comprehensive figures and textual descriptions, clear structure, and detailed experiments.
2.	Starting from the weaknesses of PEARL—including its rollback-oblivious nature, static draft length, and inadequate handling of pre-verify/post-verify rollbacks—the authors explain the rationale behind proposing the Hybrid Rollback-Aware Draft Structure (H-RAD). They further validate the design of H-RAD as a three-class classification framework through experiments, and combine it with in-depth theoretical analyses (e.g., Theorem 1 on latency under rollback, truncated geometric distribution of accepted tokens) to develop the SpecBranch method. This approach achieves excellent speedup ratios among training-free speculative sampling methods.
3.	Detailed comparative analyses are conducted on hyperparameters (e.g., maximum branch number kmax, confidence threshold ϵ, feature layer count K) and core components (H-RAD, branch resampling) that influence the speedup ratio.
4.	The paper’s analysis of the Rollback Rate (RB) clearly identifies the time-consuming bottlenecks of speculative decoding (SD) methods, and effectively demonstrates the superiority of SpecBranch in reducing rollback tokens (up to 50% reduction for poorly aligned model pairs).

**Weaknesses:**

1. There are several errors in the paper, such as the incorrect labeling of "Lookahead" in Figure 1(c) and the syntax issue with "foreach" at Line 668.
2. The superiority of the proposed method lies in its training-free characteristic, yet its acceleration performance is not as excellent as that of training-required methods like EAGLE-3. Furthermore, the so-called "training-free" essentially relies on pre-existing draft-target model pairs. Additionally, during inference, this parallel training-free method consumes more GPU resources than EAGLE and poses greater challenges in deployment. Therefore, it is anticipated that the authors will provide deployment implementations on frameworks such as vLLM or integrate SpecBranch into EAGLE, delivering more practical solutions to the industry.

**Questions:**

The authors have conducted numerous analytical experiments, and the paper is quite complexly written. However, I believe the method in the paper is actually very simple—it merely adds an MLP on top of PEARL for judgment. I need the authors to explain the timing sequence of the entire process, which should be described most clearly in the main text, especially how the subsequent tokens of the branches are generated and verified.

---

> ### Author Response · Authors · 2025-11-19
> **Response to Reviewer  MCa5 (Part 1 / 4)**
>
> We really appreciate the reviewer for the detailed and constructive feedback. We are pleased to see that you found our proposed integration of branch prediction into speculative decoding to be a meaningful and practically impactful direction.
>
> We believe our response has addressed most of your concerns and we respectfully ask **if you're comfortable to re-consider the rating in light of our efforts and dedication.** Below, we provide detailed responses to all the comments:
>
> ***
>
> ***Q1：There are several errors in the paper, such as the incorrect labeling of "Lookahead" in Figure 1(c) and the syntax issue with "foreach" at Line 668.***
>
> A1: Thanks for the catch. We will correct these typos and conduct a complete check in the camera-ready version.
>
> ***
> ***Q2： Additionally, during inference, this parallel training-free method consumes more GPU resources than EAGLE and poses greater challenges in deployment.***
>
> A2: Thanks for the good question about system implementation.  Actually, SpecBranch does not consume more GPU resources than EAGLE. We've discussed memory-constrained scenarios in **Appendix G.1** and are willing to share our thoughts with the Reviewer. Our main experiments assume memory-abundant settings with separate GPUs for the draft and target models (NVLink), making SpecBranch ideal for multi-GPU or even resource-constrained cloud-edge environments in practice. It also works well in heterogeneous accelerator setups, avoiding memory contention by deploying models across different accelerators.
>
> **For memory-constrained scenarios using only one A100 40GB GPU**, we employ pipeline parallelism (PP) to optimize resource usage. This allows the target model to run sequentially across GPUs while the draft model operates in parallel. Experiments with Deepseek 1.3B & 33B on Spec-Bench benchmark show that SpecBranch maintains **90%** of its performance, outperforming vanilla SD and auto-regressive decoding in **Appendix G.1.** We think that there is more room for system-algorithm co-design in the future by leveraging new principles of parallel decoding.

---

> ### Author Response · Authors · 2025-11-19
> **Response to Reviewer  MCa5 (Part 2 / 4)**
>
> ***Q3：The superiority of the proposed method lies in its training-free characteristic, yet its acceleration performance is not as excellent as that of training-required methods like EAGLE-3. Furthermore, the so-called "training-free" essentially relies on pre-existing draft-target model pairs.***
>
> A3: Thanks for the question. We'd like to clarify that a detailed discussion including comparisons with EAGLE\[1] and EAGLE2\[2] is available in **Appendix F.1**. To cover EAGLE-3, we have conducted more experiments:
>
> **Compared to Training-based Tree methods:**
>
> As you've pointed out, it is unfair to compare directly with EAGLE. In speculative decoding literature, training-free typically means that no additional training or fine-tuning of the draft or target LLM is required.
> Methods like EAGLE train extra decoding heads or draft models. EAGLE requires 1–2 days of training on 8 RTX 3090 GPUs for LLaMA-33B, using a dataset of 70k dialogues from ShareGPT. Moreover, EAGLE3 incurs significantly higher costs for both model training and dataset preparation.
>
> In contrast, SpecBranch uses the same pretrained draft–target pair as prior SD work -- the only training required is an MLP on feature representations. It only requires a few minutes of training on a specific dataset with a single NVIDIA A100 GPU, with nearly **250x** lower cost than EAGLE. As R2-Table1 indicates, SpecBranch achieves results similar to (and often better than) those of EAGLE families, while incurring orders of magnitude lower training cost.
>
> To further highlight SpecBranch's exceptional latency performance, we compare it with the training-required tree methods on LLaMA 2 7B and 70B, as shown in R5-Table 1.
>
> > R2-Table1: the comparison to training-required tree methods on LLaMA 2 7B and 70B
>
> | **Methods**   | **EAGLE** |         | **EAGLE-2** |         | **SpecBranch**  |           |
> | ------------- | --------- | ------- | ----------- | ------- | --------------- | --------- |
> |               | M         | Speedup | M           | Speedup | M               | Speedup   |
> | **HumanEval** | 4.45      | 3.51x   | 5.46        |  3.78x  | **18.27** | **3.59x** |
> | **GSM8k**     |  3.97     |  3.09x  | 4.49        | 3.52x   | **15.89** | **3.37x** |
> | **CNN/DM**    | 3.78      | 2.98x   |  4.98       |  3.48X  | **14.02&** | **3.23x** |
>
> Since EAGLE3 does not support LLaMA2, we conduct the comparative experiments on Qwen3-32B. Both implementations are based on the Transformers architecture.
>
> > R2-Table2: the comparison to EAGLE3 on Qwen3 0.6B and 32B
>
> | **Methods**   | **EAGLE3** |         | **SpecBranch** |           |
> | ------------- | ---------- | ------- | -------------- | --------- |
> |               | M          | Speedup | M              | Speedup   |
> | **HumanEval** | 2.88       | 2.16x   | **5.62**       | **2.57x** |
> | **GSM8k**     |  3.17      |  2.36x  | **7.64**       | **2.69x** |
> | **CNN/DM**    | 2.63       | 2.07x   | **4.96**       | **2.35x** |
>
> As shown in the table, SpecBranch achieves an impressive 3.59x speedup, outperforming EAGLE (3.51x) while trailing slightly behind EAGLE2 (3.78x). **Notably, EAGLE3 yields limited acceleration on the Qwen3 model, with SpecBranch consistently outperforming it across all datasets.** These results validate the significant potential of our branch-parallel framework and the H-RAD predictor. We recognize that a key design of EAGLE3 is the use of a dedicated draft model, which is orthogonal to our work. SpecBranch could also leverage this to conduct on-policy online distillation. This is an independent work that we aim to develop specialized draft models for parallel SD in the future.
>
> **Compared to Training-Free  methods:**
>
> We have also included recent model training-free methods, such as REST \[4] (retrieval tree structure self-SD), Ouroboros \[5] (lookahead tree structure), and Swift \[6] (layer-skip tree structure self-SD), evaluated on SpecBench using LLaMA 2 7B and 70B.
>
> > R2-Table3: the comparison to training-free  tree methods
>
> | **Methods**   | **REST** |         | **Ouroboros** |         | **SWIFT** |          | **SpecBranch**  |           |
> | ------------- | -------- | ------- | ------------- | ------- | --------- | -------- | --------------- | --------- |
> |               | M        | Speedup | M             | Speedup | M         |  Speedup | M               | Speedup   |
> | **HumanEval** | 1.97     | 1.78x   | 5.16          | 2.10x   | 4.27      | 1.56x    | **18.27**       | **3.59x** |
> | **GSM8k**     |  1.65    | 1.46x   | 5.96          |  2.58x  | 2.99      | 1.43x    | **15.89** | **3.37x** |
> | **CNN/DM**    | 1.68     | 1.47x   |  3.28         | 1.57x   | 3.87      | 1.45x    | **14.02** | **3.23x** |
>
> The comparisons in the table above show that SpecBranch outperforms other tree-structure methods. Moreover, the parallel framework can be orthogonally combined with these methods, offering significant future potential.
>
> ***

---

> ### Author Response · Authors · 2025-11-19
> **Response to Reviewer  MCa5 (Part 3 / 4)**
>
> ***Q4：Therefore, it is anticipated that the authors will provide deployment implementations on frameworks such as vLLM or integrate SpecBranch into EAGLE, delivering more practical solutions to the industry.***
>
> A4: This is an encouraging comment! In fact, this work is in progress to be merged into the recently proposed nano-PEARL\[7] framework, a lightweight, high-concurrency inference engine based on **nano-vllm** that aims for industrial settings. We're also reaching our SpecBranch to industrial partners for pilot studies in their production environments.
>
> As another reviewer pointed out, an important attribute is the high-batch scenario, and we have conducted new high-batch performance comparisons between our specbranch and EAGLE3 on the Qwen3-32B model, as shown in R2-Table4.
>
> > R2-Table4:  High-batch comparison with EAGLE3  on qwen3-32B.
>
> | **HumanEval**  | **bs=1**  | **bs=4**  | **bs=8**  | **bs=16** | **bs=32** |
> | -------------- | --------- | --------- | --------- | --------- | --------- |
> | EAGLE3         | 1.98x     | 1.67x     | 1.34x     | 1.13x     | 1.05x     |
> | **SpecBranch** | **2.34x** | **2.11x** | **1.97x** | **1.74x** | **1.55x** |
> | **GSM8K**      | **bs=1**  | **bs=4**  | **bs=8**  | **bs=16** | **bs=32** |
> | EAGLE3         | 2.03x     | 1.74x     | 1.43x     | 1.15x     | 1.07x     |
> | **SpecBranch** | **2.57x** | **2.34x** | **2.06x** | **1.88x** | **1.63x** |
>
> The results indicate that EAGLE3's performance degrades under high concurrency. This stems from its tree-based design, where computational load scales with tree size $\times$ batch size, rapidly saturating hardware. In contrast, SpecBranch mitigates this by selectively branching only on low-confidence tokens. Using a reduced $k\_{max}$, we limit the branch count and effectively delay compute saturation. Furthermore, SpecBranch employs a lightweight verification step. This bypasses the complex tree attention masks that bottleneck tree-based methods at large batch sizes. Consequently, SpecBranch demonstrates superior robustness in high-throughput environments. We will explore on-policy online distillation and **integration into frameworks like vLLM and Sglang in the future.**
>
> In addition, we are **exploring further integration with EAGLE**. For example, we can train a 4-layer EAGLE model with early-exit capability, enabling dynamic adjustment of the draft model’s layers based on context, using parameters from {1, 2, 3, 4} layers. By combining this early-exit, multi-layer draft model with the target model for parallel prediction, we can form a new branching parallel framework, which is an exciting direction.

---

> > ### Author Response · Authors · 2025-11-19
> > **Response to Reviewer  MCa5 (Part 4 / 4)**
> >
> > ***Q5： However, I believe the method in the paper is actually very simple—it merely adds an MLP on top of PEARL for judgment. I need the authors to explain the timing sequence of the entire process, which should be described most clearly in the main text, especially how the subsequent tokens of the branches are generated and verified.***
> >
> > A5: We thank the reviewer for the comment. As an old saying goes: "Simplicity is the ultimate sophistication --- Da Vinci". Although it may appear to the reviewer as “PEARL + an MLP” on the surface, SpecBranch actually differs from a simple method. The MLP/H-RAD is a synergistic, subtle co-design into the parallel framework with rollback awareness, whereas token rollbacks are neglected in the original PEARL design.
> >
> > First, the MLP/H-RAD design does not simply **judge** tokens, but **controls when and where to branch and how many branches to spawn**, which in turn demands a re-design of the entire draft–verify pipeline. If you actually look into the design pipeline of parallelism, it is quite different from PEARL, though they're under the same parallelism principle.
> >
> > The second difference is the explicit consideration of rollback awareness, which allows us to branch only at high-uncertainty points, verify branch points (not entire trees), and immediately stop sponging doomed branches once verification fails. This is why SpecBranch can reduce rollback tokens by up to 50% and achieve higher speedups than PEARL, unlike a simple method. We'd definitely follow the reviewer's suggestions to make the writing clearer.
> >
> > A more **detailed description of our method** is available as follows:
> >
> > **(1) Drafting phase.** Given a prefix X₁:ₜ₋₁, the draft model generates a sequence of candidate tokens. H-RAD takes multi-layer target features (cached from the previous step) and classifies the current position into three cases: (i) all-accept, (ii) all-reject, or (iii) confidence-based “soft” region.
> >
> > **(2) Branch point selection.** If we fall into the soft region, H-RAD marks the current token $x\_b$ as a branch point. At this point, we split the draft: the prefix X₁:ᵦ₋₁ is sent to the target model for verification, while the draft model samples top-k alternatives for $x\_b$ and continues drafting k parallel branches, all sharing the same KV-cache for X₁:ᵦ₋₁.
> >
> > **(3) Verification phase.** In parallel with branch drafting, the target model verifies the previous draft segment X₁:ᵦ₋₁. If any token in this segment is rejected, all pending branches are discarded and we fall back to the standard SD resampling step, just like in PEARL.
> >
> > **(4) Branch verification and selection**. If X₁:ᵦ₋₁ is fully accepted, the target model computes the logits $p(x\_b)$ at the branch point. We then apply a **branch speculative sampling  (detailed in Appendix D)** to the k branch candidates {$x\_b^i$} using both p and q to ensure an identical sampling distribution. As soon as one candidate is accepted, we keep the corresponding branch and drop all others (and their KV-caches). In other words, the target model only ever continues with a **single** sequence, so no tree attention is needed.
> >
> > **(5) Posterior drafting.** For the retained branch, we now have updated target features. H-RAD is invoked again to decide the next draft length and potential branch point, and the process repeats.
> >
> > A detailed profiling example is provided in **Appendix C.**
> >
> >
> > \[1] EAGLE: Speculative Sampling Requires Rethinking Feature Uncertainty. Li et.al. ICML 2024.
> >
> > \[2] EAGLE-2: Faster Inference of Language Models with Dynamic Draft Trees. Li et.al. EMNLP 2024.
> >
> > \[3] EAGLE-3: Scaling up Inference Acceleration of Large Language Models via Training-Time Test. Li et.al NIPS 202
> >
> > \[4] REST: Retrieval-based speculative decoding. He et.al. ACL 2024.
> >
> > \[5] Ouroboros: Generating Longer Drafts Phrase by Phrase for Faster Speculative Decoding. Zhao et.al. EMNLP2024
> >
> > \[6] SWIFT: On-the-Fly Self-Speculative Decoding for LLM Inference Acceleration. Xia et.al. ICLR 2025
> >
> > \[7] Parallel speculative decoding with adaptive draft length. Liu et.al. ICLR 2025
> >
> > ***
> >
> > **We are happy to see from your comments that you have an overall positive view of our work, and we respectfully ask if you'd be comfortable to adjust your ratings,&#x20;**&#x62;ased on all the new experiments and justification to your questions. Thank you once again for your time and effort. We look forward to incorporating these insightful discussions into the final camera-ready version.

---

> ### Author Response · Authors · 2025-11-25
>
> Dear Reviewer,
>
> I hope this message finds you well. As the discussion period is nearing its end, I wanted to ensure we have addressed all your concerns satisfactorily. If there are any additional points or feedback you'd like us to consider, please let us know. Your insights are invaluable to us, and we’re eager to address any remaining issues to improve our work.
>
> Thank you for your time and effort in reviewing our paper!

---

### Official Review · Reviewer_him1 · 2025-11-01

**Soundness:** 4
**Presentation:** 4
**Contribution:** 3
**Rating:** 6
**Confidence:** 4

**Summary:**

This paper seeks to address the pipeline bubble that occurs during the verification step in speculative decoding. While previous research has looked at speculating in parallel to performing verification, they have done so by making simplistic assumptions about draft acceptance length. SpecBranch on the other hand uses a hybrid predictor to intelligently decide how to spend parallel speculation resources. The SpecBranch method leads to substantial speed ups on multiple benchmarks.

**Strengths:**

* The writing and figures used throughout the paper are very clean.
* The theoretical model of parallel speculative decoding is well motivated and provides a good model for understanding tradeoffs.
* The idea of rollback aware branched parallelism is novel and addresses a large gap with current speculative decoding methods.
* The empirical analysis presented is robust as the method is tested across a large range of tasks and across multiple model sizes and model families. Furthermore, the speedup that was achieved is substantial compared to most improvements to speculative decoding.

**Weaknesses:**

* The largest weakness is that the analysis is only presented for the batch size = 1 setting. While this makes sense as the low batch setting is most relevant for speculative decoding methods, it would be helpful to profile the scaling behavior as the batch size is increased.

**Questions:**

None right now.

---

> ### Author Response · Authors · 2025-11-19
> **Response to Reviewer  him1 (Part 1 / 2)**
>
> We deeply appreciate Reviewer him1 for the positive and insightful feedback. We are equally encouraged by your recognition of the novelty of the proposed “rollback aware branched parallelism”. Your overall evaluation means a lot to researchers "**like a lantern in the dark**" that motivates us to continue refining and advancing this line of research.
>
> Your sharp catch on the high batch scenario is truly a meaningful and insightful one, which is an ongoing challenge for SD in general. To assess performance under these scenarios, we conduct additional experiments.
>
> ***
>
> ***Q1：The largest weakness is that the analysis is only presented for the batch size = 1 setting. While this makes sense as the low batch setting is most relevant for speculative decoding methods, it would be helpful to profile the scaling behavior as the batch size is increased.***
>
> A1: First, we'd like to point out that the majority of mainstream speculative decoding (SD) research is conducted and benchmarked under batch size=1 - our main results intend to align with their evaluation settings. Well, indeed, SD algorithms often struggle as batch sizes increase in industrial settings. The fundamental challenge that may nullify its gain comes from the tension between computation/data transfer: SD typically leverages additional computation to alleviate memory-bound operations (decoding). However, in high-batch scenarios, the system rapidly **becomes compute-bound.** This may adversely diminish the acceleration gains of SD, sometimes even causing it to underperform the vanilla autoregressive (AR) decoding. This limitation is present for all SOTA methods. For instance, EAGLE3\[1] (sglang) confirms that its acceleration advantage becomes negligible at batch sizes greater than $8$.
>
> **Implementation.** To address the reviewer's concerns, we extend our work to consider high-batch scenarios. We leverage the recently proposed nano-PEARL\[2] framework, a lightweight, high-concurrency inference engine based on nano-vllm. We integrate SpecBranch into this framework to enable efficient high-batch inference, fully supporting modern acceleration methods such as page attention, flash attention, tensor parallelism (TP), and CUDA graphs.
>
> The experiments are conducted on 3 NVIDIA A100-80G (NVLINK) GPUs. For SpecBranch, we apply tensor parallelism (TP) with TP=1 for the draft model (Qwen3-0.6B), TP=2 for the target model and set $k\_{max}=3$ for this high-concurrency environment, keeping other settings consistent with our main experiments. The EAGLE3 baseline is implemented with TP=2, adhering to its official configuration. The autoregressive (AR) baselines for each method are also run within their respective frameworks using TP=2, and CUDA graphs are enabled for all experimental runs.

---

> ### Author Response · Authors · 2025-11-19
> **Response to Reviewer  him1 (Part 2 / 2)**
>
> **New Experiments.** The results on high-batch performance comparisons between SpecBranch and EAGLE3 on the Qwen3-32B model are shown in R1-Table1.
>
> > R1-Table1:  High-batch comparison with EAGLE3  on Qwen3-32B.
>
> | **HumanEval**  | **bs=1**  | **bs=4**  | **bs=8**  | **bs=16** | **bs=32** |
> | -------------- | --------- | --------- | --------- | --------- | --------- |
> | EAGLE3         | 1.98x     | 1.67x     | 1.34x     | 1.13x     | 1.05x     |
> | **SpecBranch** | **2.34x** | **2.11x** | **1.97x** | **1.74x** | **1.55x** |
> | **GSM8K**      | **bs=1**  | **bs=4**  | **bs=8**  | **bs=16** | **bs=32** |
> | EAGLE3         | 2.03x     | 1.74x     | 1.43x     | 1.15x     | 1.07x     |
> | **SpecBranch** | **2.57x** | **2.34x** | **2.06x** | **1.88x** | **1.63x** |
>
> **Results Analysis**. We can see that EAGLE3’s speedup reduces significantly as batch size increases, **particularly in high-concurrency scenarios (only 1.07x under bs=32)**. This limitation originates from EAGLE3’s tree-based SD design. In practice, the effective computational load for these methods scales multiplicatively with the tree size and the batch size. As the batch size increases, the system rapidly becomes compute-bound.
>
> **1) Decoding.** In contrast, **SpecBranch maintains ≥1.55× speedup even at batch=32, while EAGLE3 nearly drops to 1.05×**. The performance gain comes from a selective strategy in SpecBranch that creates parallel branches *only* for low-confidence tokens. This method actively controls the total number of branches with much lower overhead, unlike tree-based structures that generate branches at almost every token node. This design is more compatible with the high-batch scenarios: we can set a smaller $k\_{max}$ to effectively manage the computational load and delay the onset of the compute-bound.
>
> **2) Verification.** Furthermore, SpecBranch possesses an additional advantage in the verification step. It only needs to validate its selective branch points, rather than using complex tree attention masks to process an entire draft tree. This distinction is vital in high-concurrency environments. As the batch size increases, the target model's verification latency grows. Consequently, the computational overhead of validating a full tree becomes an increasingly severe bottleneck for tree-based methods. SpecBranch's targeted verification approach generally avoids this issue.
>
> This experiment highlights SpecBranch's **robustness and superior acceleration in high-throughput environments**. We intend to build on this foundation for future optimizations. We are also actively addressing other limitations in parallel SD, such as seeking a high-quality, well-matched draft model. **We believe that parallel SD frameworks offer a competitive approach to meet the high-concurrency demands of industry-scale deployment.**
>
> \[1] EAGLE-3: Scaling up Inference Acceleration of Large Language Models via Training-Time Test. Li et.al NIPS 2025
>
> \[2] Parallel speculative decoding with adaptive draft length. Liu et.al. ICLR 2025
>
> ***
>
> We hope the response has addressed your concerns. If you have any additional concerns or comments that we may have missed in our responses, we would be most grateful for any further feedback from you to help us further enhance our work. We are more than happy to include all these discussions in the camera-ready version of this work.

---

> > ### Author Response · Authors · 2025-11-25
> >
> > Dear Reviewer,
> >
> > I hope this message finds you well. As the discussion period is nearing its end, I wanted to ensure we have addressed all your concerns satisfactorily. If there are any additional points or feedback you'd like us to consider, please let us know. Your insights are invaluable to us, and we’re eager to address any remaining issues to improve our work.
> >
> > Thank you for your time and effort in reviewing our paper!

---

### Official Review · Reviewer_mdwL · 2025-11-09

[review text omitted: it was posted to a different submission]

---

> ### Author Response · Authors · 2025-11-12
> **Mismatch in Official Review**
>
> Dear Reviewer mdwL,
>
> We hope this message finds you well. Thank you very much for your time and the valuable feedback you've provided to the ICLR community.
>
> We are writing to you to kindly raise a small issue regarding our Submission 12802. We believe that, perhaps due to a small oversight or a system bug, your review comments may have been intended for a different paper. We have carefully read your review and the insightful comments on the "RepSpec" paper mentioned within it. The feedback you provided for that work is clearly very professional and valuable, and we sincerely believe you are a very responsible and kind reviewer.
>
> However, our submission is titled "SpecBranch." We are very hopeful that you might be willing to take a second look at the "Official Review" and provide some of your valuable suggestions for our work as well.
>
> Your insights would be invaluable to us, like a lantern in the dark, and we are eager to improve our paper based on any feedback you might have.
>
> Thank you again for your time and understanding.
>
> Best regards, Authors of Submission 12802

---

> > ### Author Response · Authors · 2025-11-15
> >
> > Dear Reviewer mdwL,
> >
> > Hope you're well. We're writing to follow up on the review mismatch issue of our Submission 12802 mentioned earlier.
> >
> > We've noticed that many other submissions are encountering the same problem, so we suspect this might be a **system glitch** on the platform. To ensure we can prepare an effective rebuttal and improve our paper (titled "SpecBranch") properly, we sincerely hope this issue can be addressed promptly.
> >
> > Thank you very much for your attention and assistance.
> >
> > Best regards,
> > Authors of Submission 12802

---

> ### Comment · Area_Chair_NLpi · 2025-11-17
> **Regarding your review for Submission 12802**
>
> Dear Reviewer mdwL,
>
> I hope this email finds you well.
>
> I am the Area Chair for Submission 12802, and I'm writing to you today regarding your review for this paper.
>
> The authors have reached out to me, suggesting that your review might have been intended for a different submission. After carefully reading your comments alongside the manuscript, I am inclined to agree that there appears to be a mismatch between the review and the content of this specific paper.
>
> We certainly understand that reviewing is a demanding task, and with a heavy workload, mix-ups can occasionally happen.
>
> With the author rebuttal period drawing to a close, and to ensure a fair review process where the authors have an opportunity to respond to feedback relevant to their work, could you please take a moment to verify this and update your review for Submission 12802 at your earliest convenience?
>
> Thank you very much for your time and your valuable contributions to the conference. Please do not hesitate to contact me if you run into any issues while updating the review.
>
> Best regards,
>
> The Area Chair for Submission 12802

---

> ### Author Response · Authors · 2025-11-22
>
> Dear Reviewer mdwL,
>
> We hope you are having a good week.
>
> Following up on our previous messages regarding the review mismatch (where the comments seem to be for "RepSpec"), we wanted to let you know that we have finalized our rebuttals for the other reviewers.
>
> We remain extremely hopeful to hear your thoughts on our submission, "SpecBranch". Your perspective is very important to us. **Our main concern is that with the remaining time shrinking, we might miss the opportunity to adequately address your specific questions or clarify any potential doubts you might have about our work.**
>
> We would be deeply grateful if you could take a look at the correct PDF and share your constructive feedback. We are fully prepared to engage in the discussion as soon as your review is updated.
>
> Thank you again for your support and understanding.
>
> Best regards, Authors of Submission 12802

---

> ### Author Response · Authors · 2025-11-25
>
> Dear Reviewer,
>
> I hope this message finds you well. As the discussion period is nearing its end, I wanted to ensure we have addressed all your concerns satisfactorily. If there are any additional points or feedback you'd like us to consider, please let us know. Your insights are invaluable to us, and we’re eager to address any remaining issues to improve our work.
>
> Thank you for your time and effort in reviewing our paper!

---

### Public Comment · ~Haocheng_Sun1 · 2025-11-17
**Result of vicuna-68M**

Dear Author,

After reading your paper, I conducted tests using the double7/vicuna-68m model on Hugging Face. However, my evaluation results on PEARL differ from yours. May I ask whether your vicuna 68M model is self-trained or publicly available? If it is open-source or accessible, could you please provide the URL to facilitate reproduction of your results?

---

> ### Author Response · Authors · 2025-11-17
> **Result of vicuna-68M**
>
> Dear Sun，
>
> We appreciate your interest in our paper. We would like to clarify that all draft models utilized in our work are open-source. The specific models and their corresponding repositories URL are listed below: (1) double7/vicuna-68m-https://huggingface.co/double7/vicuna-68m ; (2) lmsys/vicuna13b-v1.3-https://huggingface.co/lmsys/vicuna-13b-v1.3. Regarding the discrepancy in PEARL results, we assure you that our findings are fully reproducible and kindly invite you to review Appendix E for the detailed experimental setup. As outlined in Appendix E.2, we evaluated standard model pairs including LLaMA, Vicuna, DeepSeek-Coder, and LLaMA-3.1, with all weights loaded in bfloat16 without quantization. Since SpecBranch is draft model training-free, no parameters were modified during evaluation. We remain available for any further discussion.
>
> Appendix E.2：To validate performance, we select state-of-the-art open-source model pairs such as the LLaMA series (JackFram /LLaMA-68M, huggyLLaMA/LLaMA-7b), Vicuna (double7/vicuna-68M, lmsys/vicuna13b-v1.3), Deepseek-Coder (deepseek-ai/deepseek-coder-1.3b-instruct, deepseek-ai/deepseek-coder33b-instruct) and LLaMA-3.1 (meta-LLaMA/LLaMA-3.1-8B-Instruct, meta-LLaMA/LLaMA-3.1-70B-Instruct) for each task. All model weights are loaded in bfloat16 format for optimized GPU inference without quantization. As a draft model training-free method, SpecBranch does not modify any draft model parameters during evaluation.
>
> Best regards, Authors of Submission 12802

---

> > ### Public Comment · ~Haocheng_Sun1 · 2025-11-17
> >
> > Thanks for your reply :-)

---

> > > ### Author Response · Authors · 2025-11-17
> > >
> > > If you have more questions, we are more than happy to assist!

---

> > ### Comment · Area_Chair_NLpi · 2025-11-17
> > **On the Public Discussion**
> >
> > To the authors and Haocheng Sun,
> >
> > Haocheng Sun: This is a good question regarding the reproducibility of the vicuna-68m model. Thank you for taking the time to verify the results.
> >
> > Authors: Thanks for the prompt and clear response. Providing the direct links and clarifying the setup details is helpful.
> >
> > This is a constructive discussion, and I will take this positive interaction into account in my final recommendation.
> >
> > Best,
> > Area Chair, Submission 12802

---

### Author Response · Authors · 2025-11-22
**General Response**

For clarity and simplicity, we will refer to Reviewers him1, MCa5, and oJJE as R1, R2, and R3, respectively, in the following response. For **Reviewer mdwl**, we remain open to any valuable feedback you may provide on SpecBranch.

We sincerely thank all reviewers for their thoughtful and constructive feedback. We are encouraged by their recognition of the key contributions and strengths of our work.

We have carefully addressed each individual comment provided by the reviewers and believe we have successfully responded to most of their concerns. **In our revised manuscript, we have incorporated the suggested experiments, additional discussions, and relevant updates to further strengthen our work.** Below, we summarize the core contributions of our study, the updates to our experiments, and the in-depth discussions included in our revision.
***
**Core Contributions of Our Work**

1. **Branch-Parallel Architecture:** We first establish theoretical models to quantify ideal parallel speculation latency and extend it towards rollback penalties in practice. Guided by these insights, we propose a branch resampling mechanism that strategically introduces parallel speculative branches to preemptively hedge against likely rejections.
2. **Hybrid Adaptive Drafting:** Based on extensive empirical analysis of adaptive draft structures, we unify the implicit (draft model confidence) and explicit (target model feature) methods into the first hybrid framework that dynamically optimizes draft lengths.
3. **Extensive Evaluation and Discussion:** We conduct extensive experiments across various models and tasks, demonstrating that SpecBranch consistently achieves a 1.8x to 4.5x speedup without draft-model training and reduces rollback tokens by 50% for poorly aligned models.
***
**Updates of experimental results during Rebuttal**

* **Main Text :** Added ablation studies regarding branch length (**R3**) and performance experiments under high-batch settings (**R1**).
* **Appendix G.9 :** Added detailed experimental results on high-batch performance comparisons with EAGLE3 (**R1**).
* **Appendix G.10 :** Detailed comparison with EAGLE3 on Qwen3-32B (**R2**).
***
**Updates of in-depth discussions during Rebuttal**

* **Main Text :** Corrected typos (**R2**) and discussed the future work and engineering integration of SpecBranch (**R2**).
* **Appendix G.7 :** Discussed the design philosophy of H-RAD and the detailed algorithmic workflow of SpecBranch (**R2**).
* **Appendix G.8 :** Discussed the relationship between draft-target model similarity and system performance (**R3**).
***
We believe these additions and clarifications comprehensively address the reviewers' concerns and enhance the overall quality of our manuscript.  All revisions are highlighted in `blue-colored` text for ease of reference. Our manuscript is updated on `Nov 22, AOE time`.

We look forward to the reviewers' favourable consideration and remain grateful for their valuable feedback.

---

### Meta-Review · Area_Chair_qXhQ · 2026-01-13

**Summary:**

Overall, reviewers found the paper technically sound and conceptually novel, highlighting the introduction of rollback-aware branch parallelism and hybrid adaptive drafting as a meaningful advance over prior speculative decoding methods. The main concerns centered on:
- Generality and practicality, including behavior under high-batch / high-throughput settings and deployment realism;
- Clarity of the method, especially the end-to-end timing sequence and how branches are generated, verified, and pruned;
- Completeness of evaluation, particularly ablations on branch length, number of branches, rollback behavior, and draft–target similarity;
- Comparison to training-based methods (e.g., EAGLE-3) and the trade-off between being “training-free” and achieving maximal speedups;
- Presentation issues, including minor typos and figure labeling errors;
- One review (mdwL) was clearly mismatched to a different paper and therefore not informative for evaluating this submission.

The rebuttal substantially expanded experiments, clarified design choices, and addressed most technical concerns. The remaining issues are primarily about presentation polish and future-work directions rather than fundamental correctness or novelty.

**Reviewer Concerns:**

Addressed by the Rebuttal

- High-batch performance (him1): The authors added extensive new experiments under high batch sizes, including direct comparisons with EAGLE-3, demonstrating that SpecBranch maintains meaningful speedups where tree-based methods degrade.

- Ablation studies (oJJE): The rebuttal includes detailed ablations on branch length, number of branches, rollback rates, and draft–target similarity across tasks and model scales, directly responding to the reviewer’s requests.

- Method clarity and timing sequence (MCa5): The authors provided a step-by-step description of drafting, branch selection, verification, rollback handling, and branch resolution, significantly improving interpretability.

- Comparison with training-based methods (MCa5): Additional quantitative comparisons and a clearer discussion of training cost vs. inference benefit were added, contextualizing SpecBranch relative to EAGLE/EAGLE-3.

- Reproducibility concerns (public discussion): Model sources, configurations, and evaluation details were clarified, and the Area Chair explicitly acknowledged the constructive resolution.

Still Outstanding / Minor

- Presentation polish: Minor typos and figure labeling issues remain to be fixed in the camera-ready.

- Deployment maturity: While the authors discussed integration with nano-PEARL/vLLM-style systems and ongoing work, fully production-ready implementations are future work rather than completed artifacts.

- Formal theoretical depth: Some theoretical explanations remain intuitive rather than fully formalized, though this was not raised as a blocking issue by most reviewers.

**Reviewer Scores:**

Reviewer mdwL: Not applicable. This review was confirmed by the Area Chair to be mismatched to a different paper and should not factor into the decision.

Reviewer him1: Likely increase from 6 → 7. The main concern (lack of high-batch analysis) was thoroughly addressed with new experiments and clear analysis.

Reviewer MCa5: Likely increase from 4 → 6 (borderline accept). Substantive concerns about clarity, comparisons, and deployment were directly addressed with detailed responses, new experiments, and clearer positioning.

Reviewer oJJE: Likely remain at 8 or slightly strengthen confidence. The requested ablations and generality analyses were comprehensively added, reinforcing an already positive assessment.

---

### Decision · Program_Chairs · 2026-01-26

Accept (Poster)